# Learning Flexible Body Collision Dynamics with Hierarchical Contact Mesh Transformer

**Youn-Yeol Yu**[1]**, Jeongwhan Choi**[1]**, Woojin Cho**[1]**, Kookjin Lee**[2]**,**
**Nayong Kim**[3]**, Kiseok Chang**[3]**, Chang-Seung Woo**[3]**, Ilho Kim**[3]**,**
**Seok-Woo Lee**[3]**, Joon-Young Yang**[3]**, Sooyoung Yoon**[3]**, Noseong Park**[4]*
[1]Yonsei University    [2]Arizona State University    [3] LG Display Co., Ltd.    [4]KAIST
{yyyou,jeongwhan.choi,snowmoon}@yonsei.ac.kr,
kookjin.lee@asu.edu, noseong@kaist.ac.kr,
{nayong.kim,kschang,wcs0916,ilho.kim,ardorsu,masteryang,syyoon}@lgdisplay.com

## Abstract

Recently, many mesh-based graph neural network (GNN) models have been proposed for modeling complex high-dimensional physical systems. Remarkable achievements have been made in significantly reducing the solving time compared to traditional numerical solvers. These methods are typically designed to i) reduce the computational cost in solving physical dynamics and/or ii) propose techniques to enhance the solution accuracy in fluid and rigid body dynamics. However, it remains under-explored whether they are effective in addressing the challenges of flexible body dynamics, where instantaneous collisions occur within a very short timeframe. In this paper, we present Hierarchical Contact Mesh Transformer (HCMT), which uses hierarchical mesh structures and can learn long-range dependencies (occurred by collisions) among spatially distant positions of a body — two close positions in a higher-level mesh correspond to two distant positions in a lower-level mesh. HCMT enables long-range interactions, and the hierarchical mesh structure quickly propagates collision effects to faraway positions. To this end, it consists of a contact mesh Transformer and a hierarchical mesh Transformer (CMT and HMT, respectively). Lastly, we propose a flexible body dynamics dataset, consisting of trajectories that reflect experimental settings frequently used in the display industry for product designs. We also compare the performance of several baselines using well-known benchmark datasets. Our results show that HCMT provides significant performance improvements over existing methods. Our code is available at `https://github.com/yuyudeep/hcmt`.

## 1 Introduction

There is escalating interest in accelerating or replacing expensive traditional numerical methods with learning-based simulators (Li et al., 2020; 2021; Raissi et al., 2019; Cho et al., 2023). Learning-based simulators have delivered promising outcomes across various domains, e.g., molecular (Noé et al., 2020), aero (Bhatnagar et al., 2019), fluid (Kochkov et al., 2021; Stachenfeld et al., 2021), and rigid body dynamics (Byravan & Fox, 2017). In particular, graph neural networks (GNNs) with a mesh have demonstrated their strength and adaptability on these topics (Sanchez-Gonzalez et al., 2020; Pfaff et al., 2020). These methods are capable of i) directly operating on simulation meshes and ii) modeling systems with intricate domain boundaries. However, solving flexible dynamics with contacts remains under-explored.

Fig. 1 and Table 1 represent the complexity of various physical systems. Flexible dynamics must consider factors such as mass, damping, and stiffness, and due to its high non-linearity, it presents challenging problems (see Appendix A for the importance of flexible dynamics with contacts). Therefore, in solving collision problems in flexible dynamics within a very short timeframe, it is

---

*Corresponding Author.

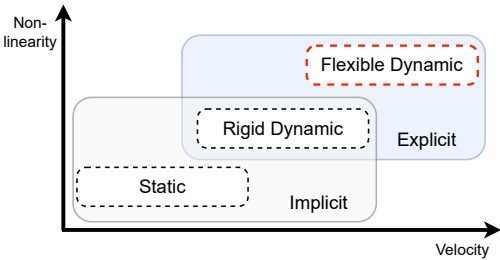 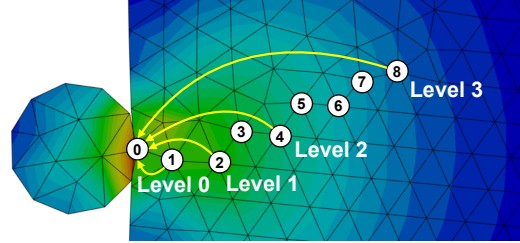

Figure 1: Relationship between velocity and non-linearity in various physical systems. Typically, implicit methods are used in static system, while explicit methods (Chang, 2007) are employed in dynamic domain. Flexible dynamics solve problems with highly non-linear characteristics and occur quickly in a very short time.

Figure 2: An illustration of the contact propagation via one-hop propagation in various levels. Level $i$ is the $i$-th level mesh (after pooling nodes from the previous level) in our hierarchical mesh structure. After node pooling, our Transformer accelerates contact propagation, which is a key in simulating flexible body collision dynamics.

Table 1: A comparison table of various systems. The biggest difference between rigid and flexible dynamics is that stress cannot be calculated. $\mathbf{M(x)}$, $\mathbf{D(x)}$, and $\mathbf{T(x)}$ represent mass, damping, and stiffness matrices, respectively. In Table, we drop the dependency on $\mathbf{x}$ for simplicity. $\mathbf{f}$ denotes an external force vector in governing equations. $\ddot{\mathbf{x}}$ and $\sigma$ are acceleration and stress, respectively.

| Behavior | System | Governing Equation | Output | Solver | Dataset |
|---|---|---|---|---|---|
| Flexible | Static | $\mathbf{Tx} = \mathbf{f}$ | $\ddot{\mathbf{x}}, \sigma$ | Implicit | Deforming, Deformable Plate |
| Rigid | Dynamic | $\mathbf{M\ddot{x}}(t) + \mathbf{D\dot{x}}(t) = \mathbf{f}(t)$ | $\ddot{\mathbf{x}}$ | Explicit | Sphere Simple |
| Flexible | Dynamic | $\mathbf{M\ddot{x}}(t) + \mathbf{D\dot{x}}(t) + \mathbf{Tx}(t) = \mathbf{f}(t)$ | $\ddot{\mathbf{x}}, \sigma$ | Explicit | Impact Plate |

essential to verify whether mesh-based GNNs are effective in handling the phenomenon, where the impact of collisions is propagated far from the point of collision between two objects. GNN models typically perform local message passing (Wu et al., 2020), which means that they cannot quickly propagate the influence of collision over long distances. To achieve such long-range propagation, multiple GNN layers are required, which increases the training/inference time. Therefore, GNN-based simulators have a trade-off relation between accuracy and training time (Bojchevski et al., 2020; Zhang et al., 2022).

Recent studies (Janny et al., 2023; Han et al., 2022) in fluid dynamics and mesh domains introduce Transformer to apply *global* self-attention. However, Transformers require a computational complexity of $\mathcal{O}(N^2)$ to generate predictions (Kreuzer et al., 2021), where $N$ is the number of nodes. Additionally, flexible dynamics require an additional instantaneous contact edge for two colliding objects to represent a collision, which slows down training. For this reason, when there are a large number of nodes, the naïve adoption of Transformers for modeling collisions still incurs high costs.

To address these challenges, we propose a novel method called Hierarchical Contact Mesh Graph Transformer (HCMT). The multi-level hierarchical structure of HCMT i) quickly propagates collisions and ii) decreases training time due to the reduced number of nodes in Level $i$, i.e., the $i$-th mesh, resulting from pooling nodes from previous level (cf. Fig. 2). HCMT with higher-level meshes focuses on long-range dynamic interactions. In addition, HCMT has two Transformers: one for contact dynamics, and the other for flexible dynamics. Finally, we introduce a novel benchmark dataset named Impact Plate. Impact Plate replicates simulations conducted in the display industry for mobile phone display rigidity assessments. We evaluate our model on three publicly available datasets (Sphere Simple (Pfaff et al., 2020), Deforming Plate (Pfaff et al., 2020), Deformable Plate (Linkerhägner et al., 2023)) and the novel Impact Plate, and our model achieves consistently the best performance.

The main contributions of this paper can be summarized as follows:

- We propose a Hierarchical Contact Mesh Transformer (HCMT), which, to the best of our knowledge, incorporates collisions into flexible body dynamics for the first time.

- We efficiently use two Transformers with different roles for flexible and contact dynamics.

- We provide an Impact Plate benchmark dataset based on explicit methods where collisions occur in a very short timeframe. Various design parameters have been applied to make it suitable for use in manufacturing.

- HCMT outperforms baseline models in static, rigid, and flexible dynamics systems.

## 2 RELATED WORK

### 2.1 GNN-BASED MODELS AND HIERARCHICAL MODELS IN PHYSICAL SYSTEMS

The prediction of complex physical systems using GNNs is an active research area in scientific machine learning (Belbute-Peres et al., 2020; Rubanova et al., 2021; Mrowca et al., 2018; Li et al., 2019; 2018). The most representative work is MGN (Pfaff et al., 2020), which uses a message passing network to learn the dynamics of physical systems. It is applied to various systems such as Lagrangian and Eulerian. Due to local processing nature of MGN, signal tends to propagate slowly through the mesh (graph). MGN has a local processing that requires one layer for one propagation step. Therefore, MGN requires many layers for long-distance propagation.

Recently, several hierarchical models have been introduced to increase the propagation radius (Gao & Ji, 2019; Fortunato et al., 2022; Han et al., 2022; Janny et al., 2023; Cao et al., 2022; Grigorev et al., 2023). Hierarchical models in these studies are divided into two types: The first type is a dual-level structure: i) GMR-Transformer-GMUS (Han et al., 2022) uses a pooling method to select pivotal nodes through uniform sampling, ii) EAGLE Transformer (Janny et al., 2023) proposes a clustering-based pooling method and shows promising performance in fluid dynamics, and iii) MS-MGN (Fortunato et al., 2022) proposes a dual-layer framework that passes messages on two different resolutions (fine and coarse resolutions) for mesh-based simulation learning. The second type has a multi-level structure: i) Cao et al. (2022) analyzes the limitations of existing pooling strategies and proposes bi-stride pooling, which uses breadth-first search (BFS) to select nodes, ii) HOOD (Grigorev et al., 2023) leverages multi-level message passing with unsupervised training to predict clothing dynamics in real-time for arbitrary types of garments and body shapes.

### 2.2 CONTACT AND COLLISION MODELS

The field dealing with contact/collision problems is primarily used in computer games (De Jong et al., 2014), animation (Hahn, 1988; Erleben, 2004), and robotics (Posa et al., 2014). A method has been proposed to use a GNN for quick motion planning in path-finding as the time required for object collision detection is significant (Yu & Gao, 2021). In mesh-based GNN models, collisions are determined by recognizing edge-to-edge or edge-to-vertex interactions close to each other (Zhu et al., 2022). Additionally, as the used node-to-node contact/collision recognition method assumes that contacts always occur at nodes, FIGNET (Allen et al., 2022b) has been proposed to introduce face-to-face (or edge-to-edge) recognition and utilize face features for training. Due to the simulation-to-real gap, there are works demonstrating the learning and prediction of contact discontinuities by GNNs using both real and simulation datasets (Allen et al., 2023). Previous studies have mainly focused on addressing contact recognition issues or improving accuracy in rigid body dynamics. However, while it is possible to determine the motion of an object in rigid body dynamics, it is not applicable in flexible dynamics because the stress distribution is unknown.

## 3 METHODLOGY

### 3.1 PROBLEM DEFINITION

Our model extends the encoder-processor-decoder framework of GNS (Sanchez-Gonzalez et al., 2020) and MGN (Pfaff et al., 2020) for solving flexible dynamics. The encoder and decoder in HCMT follow those in GNS and MGN, but we enhance the processor by designing a hierarchical

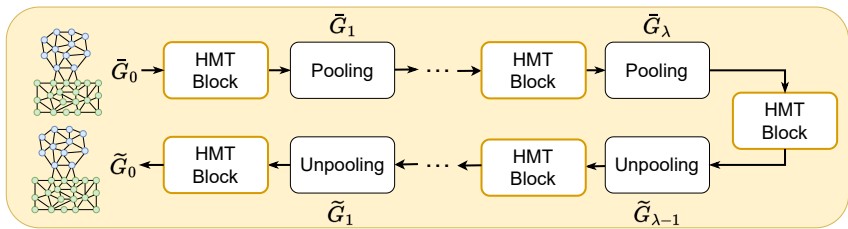

Figure 3: Overview of Hierarchical Contact Mesh Transformer (HCMT) with four layers: encoder, CMT, HMT, and decoder. The light blue graph in $G_0$ corresponds to a ball in the Impact Plate dataset, while the green graph represents a plate.

Figure 4: Hierarchical architecture of HMT layer in Fig. 3. The overall process of HMT layer is constructed by repetitively stacking HMT blocks and pooling. Utilizing this hierarchical structure is highly effective in reducing computational complexity.

Transformer that consists of i) contact mesh Transformer (CMT) for propagating contact messages and ii) hierarchical mesh Transformer (HMT) for handling long-range interactions.

At time $t$, we represent the system's state using a current mesh $M^t = (V, E)$, which includes node coordinates, density $\rho_i$, and Young's modulus $Y_i$ within each node, as well as connecting edges $m_{ij}$. The objective of HCMT is to predict the next mesh $\hat{M}^{t+1}$, utilizing both the current mesh $M^t$ and the previous meshes $M^{t-p}, \ldots, M^{t-1}$, where $p$ is a history size. Finally, The rollout trajectory can be generated through the simulator iteratively based on the previous prediction; $M^t, \hat{M}^{t+1}, \ldots, \hat{M}^{t+\kappa}$, where $\kappa$ is the length of total steps.

## 3.2 OVERALL ARCHITECTURE

Fig. 3 shows the overall architecture of HCMT which consists of an encoder, a contact mesh Transformer (CMT), a hierarchical mesh Transformer (HMT), and a decoder. The overall workflow is as follows — for simplicity but without loss of generality, we discuss predicting $\hat{M}^{t+1}$ from $M^t$ only:

1. The mesh $M^t$ is transformed into a graph $G_0 = (V_0, E_0, C)$ via encoders: the features of the $i$-th node, the $(i, j)$ mesh edge , and the $(i, q)$ contact edge, $\mathbf{f}_i$, $\mathbf{m}_{ij}$, and $\mathbf{c}_{iq}$ in $M^t$, are transformed to hidden vectors, $\mathbf{z}_i \in V_0$, $\mathbf{e}_{ij} \in E_0$, and $\mathbf{s}_{iq} \in C$, by their respective encoders.

2. CMT layer captures the contact dynamics from those transformed hidden vectors. CMT propagates contact messages through the contact edges between two colliding objects (e.g., a ball and a plate).

3. HMT layer follows CMT layer by taking the output of CMT as its input. HMT module uses a hierarchical graph with nodes properly pooled into the mesh structure to enable long-range propagation. We only consider propagation over mesh edges by a *contact-then-propagate* strategy, i.e., contact edges are excluded in this process.

4. The decoder predicts the next velocity of each node by using the last hidden representation produced by HMT layer and utilizes an updater to calculate the node positions of the objects.

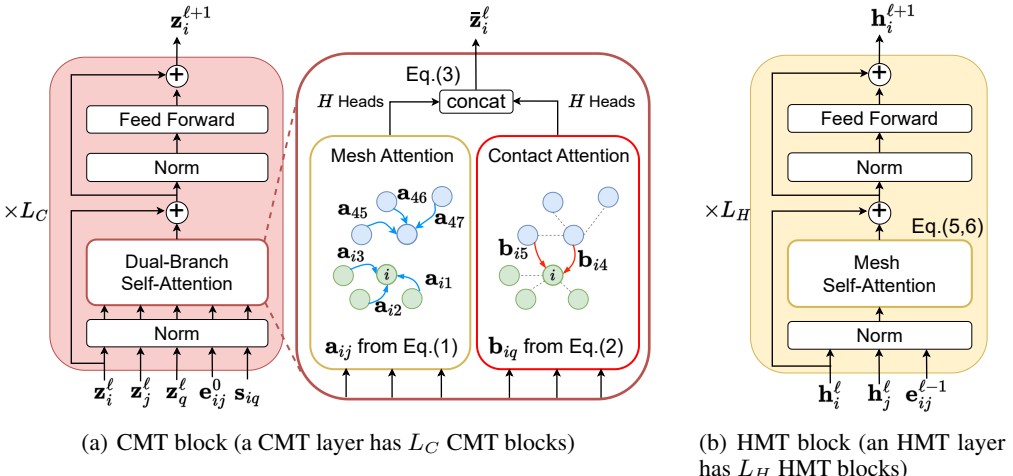

(a) CMT block (a CMT layer has $L_C$ CMT blocks)

(b) HMT block (an HMT layer has $L_H$ HMT blocks)

Figure 5: (a) depicts our proposed CMT block and dual self-attention blocks. The mesh attention on the left only computes attention weights $\mathbf{a}_{ij}$ for each of the edges of two colliding objects nodes. The contact attention on the right only considers the weights $\mathbf{b}_{iq}$ of contact edges. (b) describes our proposed HMT module. The mesh self-attention in HMT is the same as mesh attention in CMT.

## 3.3 LEARNING FLEXIBLE BODY DYNAMICS

**Features and Encoder Layer** The mesh $M^t$ is transformed into a graph $G_0 = (V_0, E_0, C)$. Mesh nodes become graph nodes $V_0$, mesh edges become bidirectional mesh-edges $E_0$ in the graph. Contact edges are connections between two different objects' nodes or self-contacts within an object, resulting in $C$ in the graph. For every node, we find all neighboring nodes within a radius and connect with them via contact edges, excluding previously connected mesh edges. The subscript '0' in $G_0$ denotes the initial level of our hierarchical mesh structure.

It is imperative that our model can learn collision behaviors in a scenario where its node positions is not fixed, and the mesh shape undergoes changes. Therefore, we structure the state of the system using two edge features: i) mesh edge feature $\mathbf{m}_{ij}^0$, where $(i, j) \in E_0$, contains information about connections within objects, and ii) contact edge feature $\mathbf{c}_{iq}$, where $(i, q) \in C$, represents connections between two objects within the collision range. Notably, the contact edge feature is a novel addition not found in fluid models. Both features are defined based on relative displacements between nodes. For detailed information of features used in each domain, see Table 7 in Appendix C.

Next, the features are transformed into a hidden vector at every node and edge. The hidden vectors for nodes are denoted as $\mathbf{z}_i$ and for mesh and contact edges as $\mathbf{e}_{ij}^0$ and $\mathbf{s}_{iq}$, respectively. The transformation is carried out by the encoder MLP $\epsilon_{\text{node}}$, $\epsilon_{\text{mesh}}$, and $\epsilon_{\text{contact}}$. The encoder $\epsilon_{\text{mesh}}$ shares weight parameters across levels because edge connections are lost when the level changes and new edge features need to be generated.

**Contact Mesh Transformer (CMT) Layer** CMT layer follows the encoder and utilizes both mesh and contact edge encoded features (see Fig. 5(a)). We use a dual-branch self-attention method with two different edges in the collision dynamics to capture the effects of the collision, resulting in a better node representation. As shown in Fig. 5(a), dual-branch self-attention branches into two self-attentions. In this case, both edge features are used to take into account the dynamically changing positions of the nodes and the length information of the edges in the dynamics system when computing the self-attention weights. For the self-attention weight $\mathbf{a}_{ij}^{k,\ell}$ for mesh edges in an object, the mesh edge feature $\mathbf{e}_{ij}^0$ can capture the relative distance between two nodes affected by collisions and is used in the calculation of the self-attention weight $\mathbf{a}_{ij}^{k,\ell}$ as shown in Equation 1. Another self-attention weight, $\mathbf{b}_{iq}^{k,\ell}$, is calculated by including the contact edge feature $\mathbf{s}_{iq}$ (see Equation 2). The two attention weights for the dual-branch self-attention of the $\ell$-th block and the $k$-th head are

defined as follows:

$$\mathbf{a}_{ij}^{k,\ell} = \text{softmax}_j \left( \text{clip}\Big( \frac{\mathbf{Q}_z^{k,\ell}\text{LN}(\mathbf{z}_i^{\ell}) \cdot \mathbf{K}_z^{k,\ell}\text{LN}(\mathbf{z}_j^{\ell})}{\sqrt{d_z}} \Big) + \mathbf{E}_z^{k,\ell}\text{LN}(\mathbf{e}_{ij}^0) \right) \cdot \mathbf{W}_z^{k,\ell}\text{LN}(\mathbf{e}_{ij}^0), \quad (1)$$

$$\mathbf{b}_{iq}^{k,\ell} = \text{softmax}_q \left( \text{clip}\Big( \frac{\mathbf{Q}_z^{k,\ell}\text{LN}(\mathbf{z}_i^{\ell}) \cdot \mathbf{K}_z^{k,\ell}\text{LN}(\mathbf{z}_q^{\ell})}{\sqrt{d_z}} \Big) + \mathbf{E}_z^{k,\ell}\text{LN}(\mathbf{s}_{iq}) \right) \cdot \mathbf{W}_z^{k,\ell}\text{LN}(\mathbf{s}_{iq}), \quad (2)$$

$$\bar{\mathbf{z}}_i^{\ell} = \text{concat}\Big( \|_{k=1}^{H} \sum_{j\in\mathcal{N}_i} \mathbf{a}_{ij}^{k,\ell}\big(\mathbf{V}_z^{k,\ell}\text{LN}(\mathbf{z}_j^{\ell})\big), \|_{k=1}^{H} \sum_{q\in\mathcal{T}_i} \mathbf{b}_{iq}^{k,\ell}\big(\mathbf{V}_z^{k}\text{LN}(\mathbf{z}_q^{\ell})\big) \Big), \quad (3)$$

where $\mathcal{N}_i$ is the set of the neighbors of node $i$, excluding contact edges between other objects and $\mathcal{T}_i$ is the set of the neighbors for contact edges between different objects. $\mathbf{a}_{ij}^{k,\ell}, \mathbf{b}_{iq}^{k,\ell}$ are the attention scores for mesh and contact edges, respectively. $\mathbf{Q}_z^{k,\ell}, \mathbf{K}_z^{k,\ell}, \mathbf{V}_z^{k,\ell}, \mathbf{E}_z^{k,\ell}, \mathbf{W}_z^{k,\ell} \in \mathbb{R}^{d_z \times d_z}$ are trainable parameters. $k = 1$ to $H$ denotes the number of attention heads, and $\|$ denotes concatenation. We use clamping for numerical stability (see Appendix I.3). $d_z$ is the dimension of node hidden vector. We adopt a Pre-Layer Norm architecture (Xiong et al., 2020), which is denoted as $\text{LN}(\cdot)$.

To move on to the next block, the output $\mathbf{z}_i^{\ell+1}$ is projected and passed to the Feed Forward Network (FFN) and represented by the residual update method as follows:

$$\mathbf{z}_i^{\ell+1} = \mathbf{z}_i^{\ell} + \mathbf{O}_z^{\ell}\bar{\mathbf{z}}_i^{\ell} + \text{FFN}_v^{\ell}(\text{LN}(\mathbf{z}_i^{\ell} + \mathbf{O}_z^{\ell}\bar{\mathbf{z}}_i^{\ell})), \quad (4)$$

where $\mathbf{O}_z^{\ell} \in \mathbb{R}^{2d_z \times 2d_z}$ is the learned output projection matrix. Notably, in our model, the edge network remains unaltered and does not undergo updates.

**Hierarchical Mesh Transformer (HMT) Layer**  Our proposed HMT layer propagates messages through mesh edges in an object. HMT layer begins after the initial collision dynamics that have been captured by CMT layer. By using only mesh edge features, HMT layer uses only the mesh self-attention, as shown in Fig. 5 (b). Note that the behavior is identical to the mesh attention in CMT layer. HMT layer uses node pooling to construct hierarchical graphs so that the propagation of nodes can reach a long-range of nodes. The updated $\bar{G}_0$ from CMT layer is pooled up to level $\lambda$ From $\bar{G}_0$. We apply the node pooling operation to construct $\{\bar{G}_1, \bar{G}_2, \cdots, \bar{G}_\lambda\}$ (with decreasing numbers of nodes) and the propagation by the mesh self-attention occurs at each level $\bar{G}_i$ (see Fig. 4). The reduced number of mesh edges due to pooling can reduce the computational complexity of the model, i.e., the training speed, and at the same time consider a longer range of interactions. HMT layer can achieve the effect of shortcut message passing, as shown in Fig. 2, and further extend the impact of collisions over long-range distances. One block of HMT layer is defined as follows:

$$\mathbf{a}_{ij}^{k,\ell} = \text{softmax}_j \left( \text{clip}\Big( \frac{\mathbf{Q}_h^{k,\ell}\text{LN}(\mathbf{h}_i^{\ell}) \cdot \mathbf{K}_h^{k,\ell}\text{LN}(\mathbf{h}_j^{\ell})}{\sqrt{d_h}} \Big) + \mathbf{E}_h^{k,\ell}\text{LN}(\mathbf{e}_{ij}^{\ell-1}) \right) \cdot \mathbf{W}_h^{k,\ell}\text{LN}(\mathbf{e}_{ij}^{\ell-1}), \quad (5)$$

$$\bar{\mathbf{h}}_i^{\ell} = \|_{k=1}^{H} \sum_{j\in\mathcal{N}_i} \mathbf{a}_{ij}^{k,\ell}(\mathbf{V}_h^{k,\ell}\text{LN}(\mathbf{h}_j^{\ell})), \quad (6)$$

$$\mathbf{h}_i^{\ell+1} = \mathbf{h}_i^{\ell} + \mathbf{O}_h^{\ell}\bar{\mathbf{h}}_i^{\ell} + \text{FFN}_h^{\ell}(\text{LN}(\mathbf{h}_i^{\ell} + \mathbf{O}_h^{\ell}\bar{\mathbf{h}}_i^{\ell})). \quad (7)$$

The $(\ell-1)$-th edge feature uses the $(L_H - \ell)$-th edge feature after reaching the final level $\lambda$. Note that $\mathbf{h}_i^1$, the input to the first block of HMT, is the same as $\mathbf{z}^{L_C}$.

**Pooling Method**  Our pooling method consists of two steps: i) sampling nodes and ii) remeshing. In the first step, For node sampling, we follow the node selection method using the breadth-first-search (BFS) proposed by Cao et al. (2022). As the level increases, the number of nodes is reduced by almost half compared to the previous level. A pooling operation is performed for each individual object.

In the second step, we regenerate mesh edges using Delaunay triangulation (Lee & Schachter, 1980; Lei et al., 2023) based on the selected nodes at each level. Delaunay triangulation produces a *well-shaped* mesh because triangles with large internal angles are selected over triangles with small internal angles when satisfying the Delaunay criterion [1]. By performing the remeshing procedure, the overall quality of the mesh is improved. For detailed results, see Tables 4 and 5 in Appendix B.

---

[1]This criterion is known as the empty circumcircle property.

**Decoder and Updater** We describe the decoder and updater based on Impact Plate dataset. According to the MGN approach, the model predicts one or more output features $\hat{\mathbf{o}}_i$ such as velocity $\hat{\dot{\mathbf{x}}}_i^t$ and next stress $\hat{\sigma}_i^{t+1}$ by employing an MLP decoder. Finally, $\hat{\dot{\mathbf{x}}}_i^t$ is used to calculate the next position $\hat{\mathbf{x}}_i^{t+1}$ through the updater, which performs a first-order integration $(\hat{\mathbf{x}}_i^{t+1} = \hat{\dot{\mathbf{x}}}_i^t + \mathbf{x}_i^t)$.

**Training Loss** We use the one-step MSE loss as a training objective. Since stresses are included, the position and stress MSE in flexible dynamics are calculated as follows:

$$\mathcal{L} = \frac{1}{|V_0|} \sum_{i=1}^{|V_0|} (\mathbf{x}_i^{t+1} - \hat{\mathbf{x}}_i^{t+1})^2 + \frac{1}{|V_0|} \sum_{i=1}^{|V_0|} (\sigma_i^{t+1} - \hat{\sigma}_i^{t+1})^2. \tag{8}$$

### 3.4 DISCUSSION

In this subsection, we discuss the importance of contact edge features in the design of HCMT and compare HCMT with existing graph Transformers in the mesh domain.

**Why Mesh and Contact Edge Features are Important?** The contact edge represents crucial information that depicts the collision phenomenon between two objects. It is defined as the relative displacement between nodes where contact occurs within a specific radius, which can distinguish the strength or weakness of a collision. When an object is deformed after a collision, the shape of the mesh (cell) also changes, so mesh edges also contain important information. Hence, collision models place considerable emphasis on considering the impacts of collisions as the decrease or increase of edge lengths. Edge consists of both mesh edge, which encompasses mesh connections within objects, and contact edge, which considers the effects of collisions for flexible dynamics.

**Comparison with Transformer-based Models** EAGLE (Janny et al., 2023), which learns fluid dynamics, does not use edge features in its Transformer but in its last GNN. In Eulerian-based fluid dynamics, node positions and edge lengths are fixed, making edge functions less useful. In flexible dynamics where contact occurs, however, contact edges carry important messages. In addition, because the object is severely deformed, mesh edge features are also greatly affected. GMR-Transformer-GMUS (Han et al., 2022) is a representative example of applying a Transformer in the mesh domain. GMR-Transformer-GMUS uses one Transformer without edge features, while our model uses two Transformers with edge features. Graph Transformer (Dwivedi & Bresson, 2020) (GT) has been extended to incorporate edge representations, which can work with information associated with edges. In comparison with them, we utilize both mesh and contact edge features to infer accurately.

## 4 EXPERIMENTS

### 4.1 EXPERIMENTAL SETTINGS

**Datasets** We use 4 datasets of 3 different types: i) Impact Plate dataset for flexible dynamics, ii) Deforming Plate and Deformable Plate for static dynamics, and iii) Sphere Simple for rigid dynamics. Impact Plate is created with traditional solver with ANSYS (Stolarski et al., 2018) to validate the efficacy of our model in flexible dynamics (see Table 6 in Appendix C).

**Baselines, Setups and Hyperparameters** As baselines, we use MGN (Pfaff et al., 2020), a state-of-the-art model in the field of complex physics, and GT (Dwivedi & Bresson, 2020) with edge features. A detailed description for all baselines is provided in Appendix D. The number of blocks $L = L_C + L_H$ is set to 15. For further details on hyperparameters, see Table 8 in Appendix E.

### 4.2 EXPERIMENTAL RESULTS

Table 2 shows the results of comparing our HCMT to two baselines. HCMT outperforms in all cases (e.g., 49% improvement in position RMSE for Impact Plate) and has the lowest standard deviations. We also report empirical complexity in Appendix F.

Table 2: RMSE (rollout-all, $\times 10^3$) for our model and the baselines. Improv. means the percentage improvement over the runner-up and **bold** denotes the best performance.

| Model | Impact Plate | | Deforming Plate | | Sphere Simple | Deformable Plate |
|---|---|---|---|---|---|---|
| | Position | Stress | Position | Stress | Position | Position |
| GT | $59.18_{\pm 4.45}$ | $39{,}291_{\pm 21{,}529}$ | $11.34_{\pm 0.28}$ | $9{,}168{,}298_{\pm 164{,}941}$ | $243.85_{\pm 141.08}$ | $13.74_{\pm 0.47}$ |
| MGN | $40.73_{\pm 2.94}$ | $35{,}871_{\pm 11{,}893}$ | $7.83_{\pm 0.16}$ | $4{,}644{,}483_{\pm 92{,}520}$ | $33.26_{\pm 6.33}$ | $10.78_{\pm 0.54}$ |
| HCMT | $\mathbf{20.71_{\pm 0.57}}$ | $\mathbf{14{,}742_{\pm 502}}$ | $\mathbf{7.49_{\pm 0.07}}$ | $\mathbf{4{,}535{,}956_{\pm 49{,}937}}$ | $\mathbf{30.41_{\pm 1.71}}$ | $\mathbf{7.67_{\pm 0.42}}$ |
| Improv. | 49.2% | 58.9% | 4.3% | 2.3% | 8.6% | 28.9% |

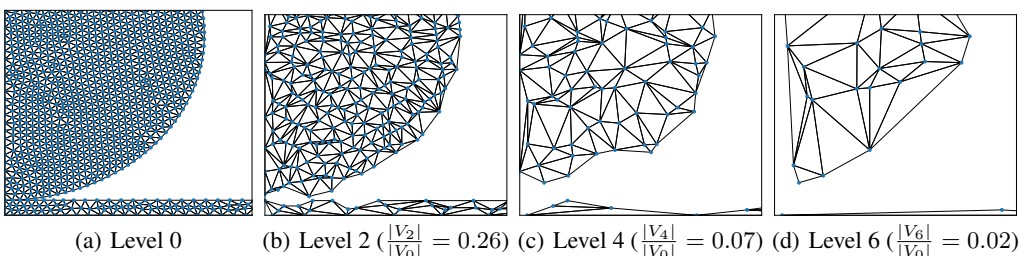

(a) Level 0    (b) Level 2 ($\frac{|V_2|}{|V_0|} = 0.26$)   (c) Level 4 ($\frac{|V_4|}{|V_0|} = 0.07$)   (d) Level 6 ($\frac{|V_6|}{|V_0|} = 0.02$)

Figure 6: Level-wise mesh visualization of Impact Plate. See more visualizations in Appendix B.

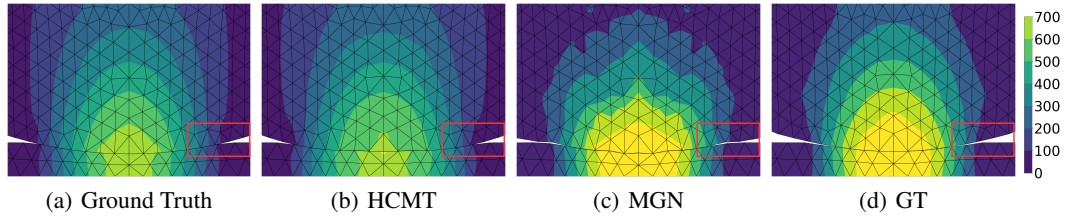

(a) Ground Truth      (b) HCMT      (c) MGN      (d) GT

Figure 7: 2D cross-sectional contour of the stress field in Impact Plate. In the red bounding box, HCMT is the most similar to the node positions in the ground truth. Brighter colors mean higher stress. Rollout images of other datasets can be found in Appendix L.

**Result** HCMT shows superior performance in terms of long-term prediction across all domains compared to MGN and GT. In particular, we can see that HCMT outperforms other methods in Impact Plate, where strong impacts occur, and that our hierarchical architecture effectively propagates collisions. Fig. 6 shows that as the level increases, shortcut message passing is possible in our method. Fig. 7 shows the stress on the falling ball and plate after a collision, visually demonstrating the superiority of HCMT. We also report generalization abilities of HCMT in Appendix G.

**Visualize Attention Map** We visualize the dual-branch self-attention maps of CMT layer on Impact Plate in Fig. 8 — we note that the self-attention is naturally sparse in $G_0$. The contact and mesh self-attention show non-overlapping attention maps and the varying importance of contact and mesh edges. The red bounding box in Fig. 8 (a) represents the importance of contact edges between the ball and the plate, and the blue box in Fig. 8 (b) signifies the self-attention among plate nodes, and the yellow box represents interactions through edges among ball nodes (see Appendix J for self-attention map of HMT and results in different datasets).

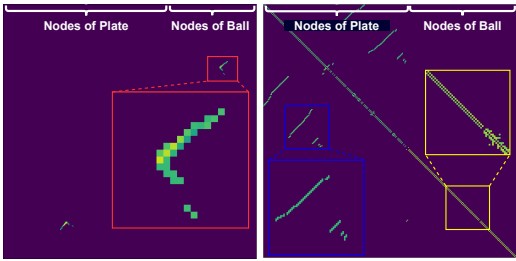

(a) Contact Self-Attention    (b) Mesh Self-Attention

Figure 8: Self-attention maps in CMT (see fig. 13 in Appendix 13 for map of HMT).

Table 3: The results of ablation studies.

| Model | Impact Plate | | Deforming Plate | | Sphere Simple | Deformable Plate |
| | Position | Stress | Position | Stress | Position | Position |
| --- | --- | --- | --- | --- | --- | --- |
| HCMT | **20.34** | **14,447** | **7.37** | **4,475,616** | **28.30** | **7.67** |
| Late-Contact | 42.90 | 22,874 | 7.74 | 4,991,179 | 38.81 | 7.97 |
| Only HMT | 46.27 | 31,932 | 22.32 | 33,773,408 | 144.73 | 24.69 |
| Only CMT | 55.63 | 28,764 | 7.96 | 4,662,353 | 30.15 | **7.67** |
| HCMT+LPE | 21.11 | 14,920 | 7.52 | 4,618,027 | 30.16 | 7.87 |

Figure 9: Sensitivity to $\lambda$. The red dashed lines represent RMSE (rollout-all, $\times 10^3$) of MGN.

## 4.3 Ablation and Sensitivity Studies

This subsection describes the ablation and sensitivity studies for HCMT. Additional studies and analyses are in Appendix I.

**Ablation Studies** We define the following 4 models for ablation studies: i) "Late-Contact", where the order of CMT layer is followed by HMT layer, ii) "Only CMT", which uses only CMT layer, iii) "Only HMT", which uses only HMT layer, and iv) "HCMT+LPE", which adds Laplacian positional encoding (LPE) (Dwivedi & Bresson, 2020). In Table 3, "Late-Contact" performs worse than HCMT. This shows that in HCMT, the *contact-then-propagate* strategy, where CMT layer is followed by HMT layer, is appropriate for propagating the forces generated by the interaction and reaction of two objects. In all cases, "Only CMT" performs better than "Only HMT". Finally, "HCMT+LPE" performs the best among the ablation models, but shows a slight performance drop compared to our HCMT. We conjecture that this is because our feature definitions already include node positions and therefore, additional positional encodings are not needed for nodes.

**Sensitivity to the Number of Level** Fig. 9 shows the position RMSE varying the number of level $\lambda$. For Impact Plate, the RMSE tends to decrease as the level increases due to short time intervals and substantial impacts. For Deformable Plate, due to a small number of nodes, there is a tendency for the RMSE to decrease as the level decreases. For Deforming Plate, levels 2 to 4 show the best RMSE because a plate is affected by an object pushing it up to about half the total plate size. For Sphere Simple, the RMSE gets worse as the level increases, but it shows the best result at level 4.

## 5 Conclusion and Future Work

We presented HCMT, a novel mesh Transformer that can effectively learn flexible body dynamics with collisions. HCMT uses a hierarchical structure and two different Transformers to enable long-range interactions and quickly propagate collision effects. We show that HCMT outperforms other baselines on a variety of systems, including flexible dynamics. We believe that HCMT has potential to be used in a variety of applications, such as product design and manufacturing. Future work topics include learnable pooling methods and combining mesh-based simulators with high-quality mesh generation models (Lei et al., 2023; Nash et al., 2020) from complex geometries for design optimization (Allen et al., 2022a).

ACKNOWLEDGMENTS

This work was supported by the LG Display and an IITP grant funded by the Korean government (MSIT) (No.2020-0-01361, Artificial Intelligence Graduate School Program (Yonsei University)).

ETHICS STATEMENT

There is no ethical problem with our experimental settings and results.

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

## A  WHY IS A STUDY ON FLEXIBLE DYNAMICS NECESSARY?

Collision simulation is a widely utilized methodology in various industries such as shipping (Goerlandt & Kujala, 2011), automotive (Gui et al., 2018), aviation (Liu et al., 2021), etc. In collision dynamics, there are i) rigid body dynamics where the shape of the mesh remains unchanged and ii) flexible body dynamics where it can change. Rigid body dynamics forms meshes of objects where collisions can occur, allowing them to detect collisions and predict the movement of the objects by calculating their accelerations. However, the goal of flexible body dynamics is to determine the maximum stress value and the part of interest on an object after a collision has occurred. Exceeding the yield strength (Pavlina & Van Tyne, 2008) results in product damages, making it possible to produce robust products through simulations for detecting vulnerable areas and changing their designs. For instance, when evaluating the rigidity of a display panel to prevent damages caused by external forces, a cover is attached to the back of the panel (Xue et al., 2013). The reliability varies based on design elements such as the material, shape, thickness, and material of the cover. Therefore, before prototype production, a robust design (Jirathearanat et al., 2004) can be derived through flexible body dynamics.

Among previous studies, the hierarchical GNN model, BSMS-GNN (Cao et al., 2022), despite performing long-range interactions, exhibits lower accuracy in the collision domain compared to MGN. GNN-based hierarchical models, MS-GNN (Fortunato et al., 2022) and HOOD (Grigorev et al., 2023), focus on studying the fluid domain or rigid body dynamics. Therefore, we need research on flexible body dynamics and verification of the effectiveness of the hierarchical structure.

## B  POOLING METHOD DETAILS

Our pooling strategy proceeds in two steps. In the first step, nodes are selected using breadth-first search (BFS), and in the second step, Delaunay triangulation is applied to regenerate the mesh in a high-quality shape for HMT. Delaunay triangulation divides a plane into triangles by connecting points on the plane in such a way that the minimum angle of the resulting triangles is maximized. Table 4 represents the number of nodes at each level after BFS pooling. In each level the number of nodes decreases by half compared to its previous level. Table 5 shows the mesh quality before and after remeshing. The quality of the meshes (cells) is evaluated using the scaled Jacobian metric (Moxey et al., 2014), and a scaled Jacobian value closer to 1 indicates a mesh with a shape closer to an equilateral triangle, signifying higher mesh quality. Angle refers to the interior angles within the triangular cell, and as the minimum angle decreases and the maximum angle increases, the shape of the mesh appears more distorted. The angle and the scaled Jacobian values are the average values across all levels within the training data. Fig. 10 shows the results of remeshing.

Table 4: The number of nodes for each level.

| Datasets | Level 0 | Level 1 | Level 2 | Level 3 | Level 4 | Level 5 | Level 6 |
|---|---|---|---|---|---|---|---|
| Impact Plate | 2208 | 1108 | 559 | 285 | 148 | 78 | 42 |
| Deforming Plate | 1276 | 658 | 344 | 183 | 98 | 55 | 32 |
| Sphere Simple | 1731 | 898 | 523 | 330 | 224 | 170 | 138 |
| Deformable Plate | 138 | 77 | 48 | 32 | 24 | 20 | 18 |

Table 5: The quality of the cells is evaluated using the scaled Jacobian, and a scaled Jacobian closer to 1 indicates a mesh with a shape closer to an equilateral triangle, i.e., high mesh quality.

| Dataset | Before remeshing | | | After remeshing | | |
|---|---|---|---|---|---|---|
| | Max Angle (°) | Min Angle (°) | Jacobian | Max Angle (°) | Min Angle (°) | Jacobian |
| Impact Plate | 120.4 | 16.7 | 0.32 | 99.4 | 27.2 | 0.51 |
| Deforming Plate | 111.4 | 21.4 | 0.41 | 98.2 | 28.3 | 0.53 |
| Sphere Simple | 110.0 | 20.4 | 0.39 | 104.6 | 23.9 | 0.45 |
| Deformable Plate | 114.8 | 21.6 | 0.41 | 104.9 | 28.4 | 0.53 |

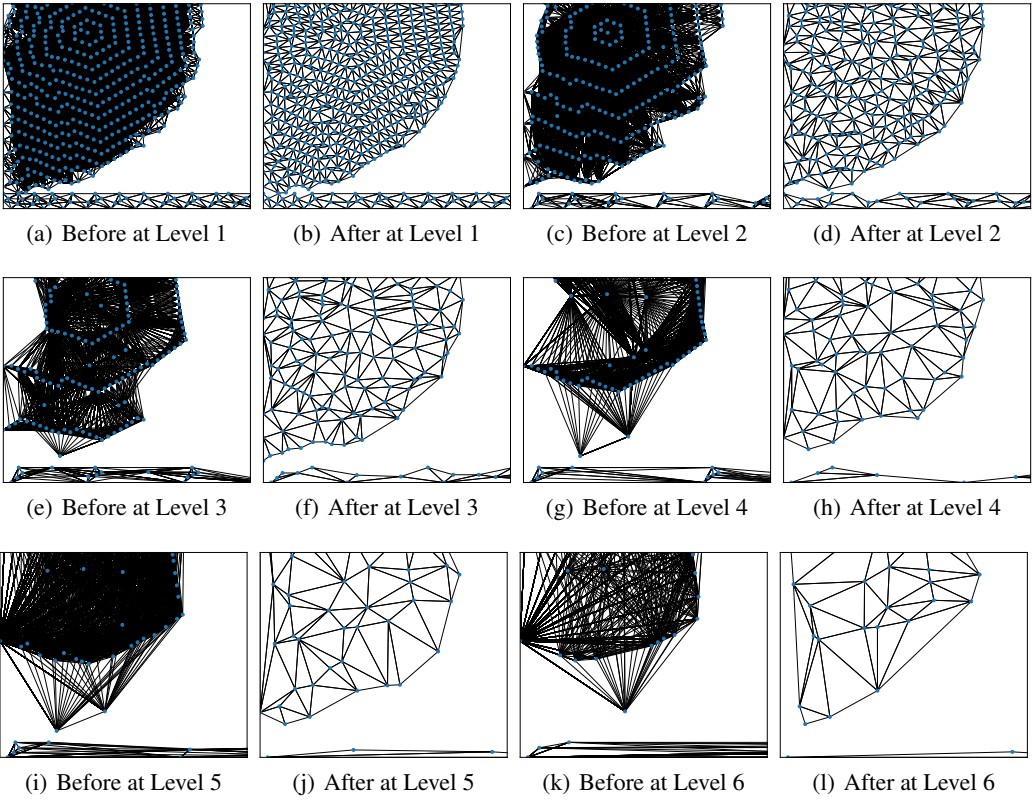



(a) Before at Level 1     (b) After at Level 1     (c) Before at Level 2     (d) After at Level 2

(e) Before at Level 3     (f) After at Level 3     (g) Before at Level 4     (h) After at Level 4

(i) Before at Level 5     (j) After at Level 5     (k) Before at Level 6     (l) After at Level 6



Figure 10: Images show before and after remeshing from the Impact Plate for each level. The blue dots represent the pooled nodes, and when Delaunay triangulation is applied, the mesh quality improves.

## C ADDITONAL DATASET DETAILS

Table 6 represents that our datasets are created with various solvers and different physics systems. Specifically, Sphere Simple involves self-contacts, while Deforming Plate, Deformable Plate, and Impact Plate do not exhibit self-contacts. In a dynamics system, there is a simulation time step $\Delta t$. The length of steps $\kappa$ is determined from the time at which the simulation ends.

Table 7 represents that the input $\mathbf{f}_i$ and output $\mathbf{o}_i$ features are defined for each domain. Node features $\mathbf{f}_i$ contain physical properties. Node types $\mathbf{n}_i$ correspond to boundary conditions. For Impact Plate, required properties such as density $\rho_i$ and Young's modulus $Y_i$ are added as node features for flexible dynamics. Young's modulus characterizes a material's ability to stretch and deform, and it is defined as the ratio of tensile stress to tensile strain.

Mesh edges are defined by a relative displacement vector $\mathbf{u}_{ij}$ and its norm $|\mathbf{u}_{ij}|$ in mesh space and a relative displacement vector $\mathbf{x}_{ij}$ and its norm $|\mathbf{x}_{ij}|$ in world space, which are used for object propagation. For every node, we find all neighboring nodes within a radius and connect with them via contact edges, excluding previously connected mesh edges. ($|\mathbf{x}_i - \mathbf{x}_q| < \gamma$). In the mesh space coordinates, the opponents between the nodes are the same in all steps in one trajectory. However, the position of the nodes represented by the world space coordinates has a different value for each step. For example, in Sphere Simple, the world space containing the position of the node where the ball or cloth moves is expressed as 3D coordinates, and the cloth and ball without thickness have a mesh space expressed as 2D coordinates.

Table 6: Dataset description: Simulators, system timeframe, etc. Self-contact indicates the case where different nodes of the same object collide with each other.

| Datasets | $\Delta t(ms)$ | Steps $\kappa$ | Simulator | Mesh Type | Self-Contact | System | Dimension |
|---|---|---|---|---|---|---|---|
| Impact Plate | 0.002 | 52 | ANSYS | Triangles | X | Flexible dynamics | 2D |
| Deforming Plate | None | 400 | COMSOL | Tetrahedral | X | Static | 3D |
| Sphere Simple | 10 | 500 | ArcSim | Triangles | O | Rigid dynamics | 3D |
| Deformable Plate | None | 52 | SOFA | Triangles | X | Static | 2D |

Table 7: Details of features for each dataset.

| Datasets | Inputs $\mathbf{m}_{ij}$ | Inputs $\mathbf{c}_{iq}$ | Inputs $\mathbf{f}_i$ | Outputs $\mathbf{o}_i$ |
|---|---|---|---|---|
| Impact Plate | $\mathbf{u}_{ij}, \|\mathbf{u}_{ij}\|, \mathbf{x}_{ij}, \|\mathbf{x}_{ij}\|$ | $\mathbf{x}_{iq}, \|\mathbf{x}_{iq}\|$ | $\mathbf{n}_i, (\mathbf{x}_i^t - \mathbf{x}_i^{t-1}), \rho_i, Y_i$ | $\dot{\mathbf{x}}_i, \sigma_i$ |
| Deforming Plate | $\mathbf{u}_{ij}, \|\mathbf{u}_{ij}\|, \mathbf{x}_{ij}, \|\mathbf{x}_{ij}\|$ | $\mathbf{x}_{iq}, \|\mathbf{x}_{iq}\|$ | $\mathbf{n}_i$ | $\dot{\mathbf{x}}_i, \sigma_i$ |
| Sphere Simple | $\mathbf{u}_{ij}, \|\mathbf{u}_{ij}\|, \mathbf{x}_{ij}, \|\mathbf{x}_{ij}\|$ | $\mathbf{x}_{iq}, \|\mathbf{x}_{iq}\|$ | $\mathbf{n}_i, (\mathbf{x}_i^t - \mathbf{x}_i^{t-1})$ | $\ddot{\mathbf{x}}_i$ |
| Deformable Plate | $\mathbf{u}_{ij}, \|\mathbf{u}_{ij}\|, \mathbf{x}_{ij}, \|\mathbf{x}_{ij}\|$ | $\mathbf{x}_{iq}, \|\mathbf{x}_{iq}\|$ | $\mathbf{n}_i, \nu_i$ | $\dot{\mathbf{x}}_i$ |

## C.1 IMPACT PLATE

The evaluation of Impact Plate follows industry-standard practices, primarily assessing the stiffness of panels — these are commonly used in the display industry (Xue et al., 2013). Impact Plate involves a strong impact in a short timeframe, representing significant geometric and contact non-linearity in its dynamics. When transitioning from 2D to 3D modeling, the computational cost increases roughly 10-20 times. Therefore, if a ball falls at the center of the plate, there is no need to use a 3D model. Fig. 11 depicts two visualizations before and after the ball impact. The most significant part is the transfer of the strong contact energy when the two objects collide. Once the falling height and the mass of ball are determined, the potential energy is calculated. At the moment of collision with the plate, the energy is converted to the internal energy of the panel, generating internal stress in the plate. If the stress value generated in the plate exceeds the ultimate strength of the material, the plate is damaged. Therefore, the final product design is determined through plate thickness and material configurations according to the required energy.

Impact Plate uses an explicit method that is typically used to calculate short-term solutions. For example, Deforming Plate represents a static system and uses implicit methods. Therefore, there is no need to predict the location of the obstacle (pusher) node because it is determined by the boundary conditions. For flexible dynamic systems, such as Impact Plate where a ball falls and collides with a place due to gravity, calculations involving the positions of the ball nodes and mesh deformations are required. Therefore, in flexible dynamic systems, the location of the obstacle (ball) must be predicted.

This benchmark dataset incorporates various design parameters, including plate properties, thickness, density, modulus, drop height, mesh size, and etc. These parameters have been randomly varied to generate a dataset consisting of 2,000 trajectories, along with 200 validation and test trajectories. We use ANSYS (Stolarski et al., 2018) to create the dataset.

## C.2 DEFORMING PLATE

Deforming Plate verifies the stress on a plate when a pusher presses on the plate. We excluded two trajectories out of 1,200 training data trajectories. Those two trajectories are excluded because they contain stress singularity (Williams, 1952), where excessive strain in a single cell causes the stress to become nearly infinite. This exclusion is necessary as it interferes with the learning process. Fig. 12 represents the trajectory number 233 in Deforming Plate. The indicated arrows represent the point with excessive deformation. At the trajectory number 173, stress singularity occurred when the pusher pressed the thin plate, so we excluded it as well.

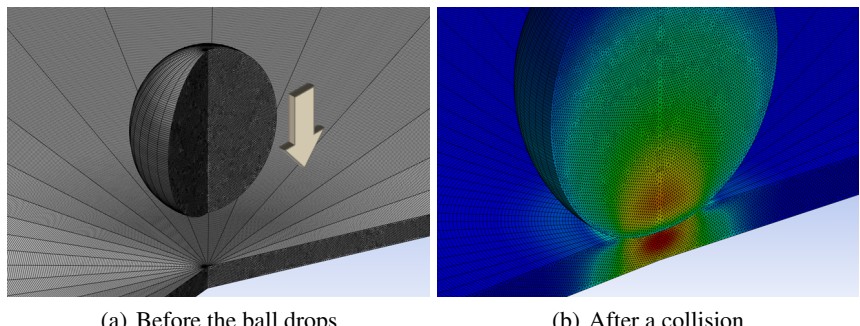

|(a) Before the ball drops|(b) After a collision|

Figure 11: After the collision, the energy from the ball is transferred to the place, resulting in the deformation of the plate and the corresponding stress distribution. The red area indicates areas of high stress.

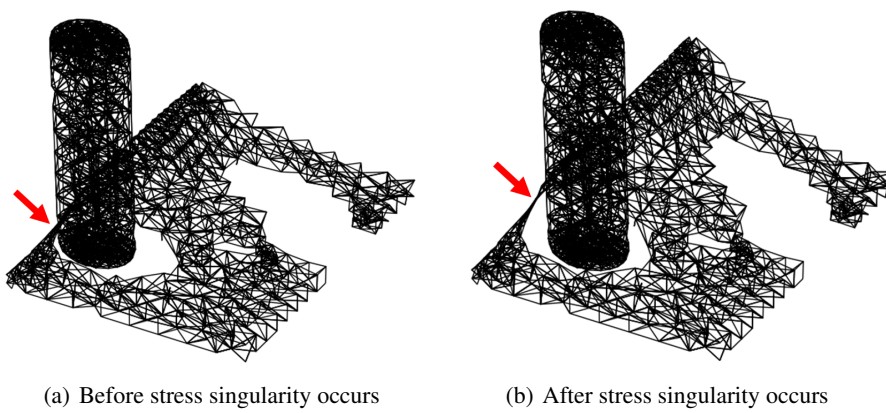

|(a) Before stress singularity occurs|(b) After stress singularity occurs|

Figure 12: Before and after stress singularity occurs due to an excessive deformation of the elements. The red arrows indicate the nodes where stress singularity has occurred.

## D    BASELINE DETAILS

We compare our methodology with two competitive approaches. The first one is MGN (Pfaff et al., 2020), which performs multiple message passing, and the second one is GT (Dwivedi & Bresson, 2020) which employs self-attention. We use the official implementation released by the authors on GitHub for all baselines:

- MGN:   `https://github.com/google-deepmind/deepmind-research/tree/master/meshgraphnets`

- GT: `https://github.com/graphdeeplearning/graphtransformer`

**MGN**   To align with the MGN methodology, we apply 15 iterations of message passing in all datasets. To learn collision phenomena, we added a contact edge encoder and processor, and directly incorporated stress values into the loss. All MLPs have a hidden vector size of 128.

**GT**   For an accurate comparison with HCMT, we use the FFN of the same HCMT without using positional encoding. The encoder/decoder were set to those in MGN. There are 15 transformer blocks with 4 heads. The hidden dimension size inside its Transformer is set to 128. FFN used three linear layers and two ReLU activations. To ensure numerical stability, the results obtained with the exponential term within the softmax function are constrained to fall in the range of $[-2, 2]$.

## E  HYPERPARAMETER DETAILS

Table 8 shows the hyperparameters in noise, radius, and the number of training steps applied to each dataset. The radius $\gamma$ is an important hyperparameter for collision detection.

Random-walk noises are added to positions in the same way as GNS (Sanchez-Gonzalez et al., 2020) and MGN (Pfaff et al., 2020) for improvements in cumulative errors. The numbers of contact propagation modules and mesh propagation modules are hyperparameters, and the number of blocks $L = L_C + L_H$ is set to 15. Following MGN, the hidden vector size of the encoder/decoder is set to 128 and the Adam optimizer is used. The batch size is set to 1, and exponential learning rate decay from $10^{-4}$ to $10^{-6}$ is applied. The hidden vector dimensions $d_z$, $d_h$ for CMT and HMT are set to 128, and the number of heads $H$ is 4. For reproducibility, we introduce the best hyperparameter configurations for each dataset in Table 8.

Table 8: best hyperparameters of configures

| Dataset | $L_C$ | $L_H$ | $\lambda$ | Noise Std. Dev. | $\gamma$ | #Training Steps |
|---|---|---|---|---|---|---|
| Impact Plate | 2 | 13 | 6 | 0.003 | 0.4 | 2,000,000 |
| Deforming Plate | 10 | 5 | 2 | 0.003 | 0.03 | 10,000,000 |
| Sphere Simple | 6 | 9 | 4 | 0.001 | 0.05 | 10,000,000 |
| Deformable Plate | 12 | 3 | 1 | 0.001 | 0.08 | 675,000 |

## F  COMPUTATIONAL EFFICIENCY

Table 9 shows Computational Efficiency. For Deforming Plate and Impact Plate, our method show similar learning times to those of MGN. Our model has a shorter training time on Sphere Simple compared to MGN. In the case of Sphere Simple, self-contacts occur and there are many contact edges. In the case of Deformable Plate, edge features are encoded at each level with a fixed number of small nodes and edges, which has a negative impact on learning time for our method.

For Impact Plate, the inference time per step for HCMT is 81 $ms$/step. For ANSYS (i.e., numerical solver), it is 14,280 $ms$/step. Compared to ANSYS, HCMT is about 176 times faster.

Table 9: Shows training time/step ($ms$) for each model.

| Model | Impact Plate | Deforming Plate | Sphere Simple | Deformable Plate |
|---|---|---|---|---|
| GT | 79.31 | 76.93 | 130.36 | 56.65 |
| MGN | 51.56 | **51.11** | 89.28 | **38.63** |
| HCMT | **51.12** | 53.53 | **59.02** | 53.16 |

## G  GENERALIZATION ABILITY OF HCMT

HCMT can generalize outside of the training distribution with respect to the design parameters of the system (e.g., plate thickness and drop height of ball) since HCMT uses the relative displacements between nodes as edge features (Sanchez-Gonzalez et al., 2020; Pfaff et al., 2020). This provides more general-purpose understanding of flexible dynamics used during training. We compare HCMT with MGN to evaluate their generalization abilities toward unseen design parameter distributions during training.

We define 3 test datasets using our original Impact Plate dataset to study the generalization ability of HCMT: i) We call the test dataset where we increase the plate thickness from 1.0 to 1.2 in the range of 0.5 to 1.0 as "Thicker Impact Plate," ii) we denote the test dataset where we increase the drop height of ball from 500 to 1000 in the range of 1000 to 1200 as "Higher Impact Plate," iii) we denote the test dataset with increased both plate thickness and ball drop height as "Thicker & Higher Impact Plate." The 3 test datasets have 50 trajectories generated from untrained design parameters.

Table 10 shows the generalization abilities of MGN and HCMT in RMSE (rollout-all, $\times 10^3$). The results of HCMT show lower RMSEs than MGN in all test datasets and show better generalization ability.

Table 10: The generalization ability of MGN and HCMT models according to various test datasets not included in the training distribution is shown in terms of RMSE (rollout-all $\times 10^3$).

| Test Datasets | MGN | | HCMT | |
|---|---|---|---|---|
| | Position | Stress | Position | Stress |
| Impact Plate | 44.18 | 28,928 | 20.34 | 14,447 |
| Thicker Impact Plate | 54.50 | 25,037 | 17.38 | 12,163 |
| Higher Impact Plate | 31.23 | 35,552 | 29.59 | 22,509 |
| Thicker & Higher Impact Plate | 25.23 | 24,719 | 21.52 | 13,745 |

## H  EFFECT OF STRESS SINGULARITY ON LEARNING

In the process of simulations using numerical solvers, singularity is a phenomenon in which the behavior of an object rapidly changes at a specific point or area within an object, resulting in very large stress. It mainly occurs when the load is concentrated in a very small area. We evaluate how stress singularities affect the learning outcomes of MGN and HCMT. The experiments are conducted with trajectories where stress singularities occur in Deforming Plate (see Appendix C.2).

Table 11 shows the results with and without stress singularities. In the case of MGN, when stress singularities are not included, its position and stress RMSEs decrease by 4.3% and 11.0%, respectively. HCMT's RMSEs decrease by 5.1% and 10.4% in the same setting. Moreover, HCMT shows better performance than MGN in all cases.

Table 11: Results of whether stress singularity is included or not in Deforming Plate (RMSE, rollout-all, $\times 10^3$).

| Model | With Singularity | | Without Singularity | |
|---|---|---|---|---|
| | Position | Stress | Position | Stress |
| MGN | 7.98 | 5,085,048 | 7.64 | 4,526,531 |
| HCMT | 7.77 | 4,995,446 | 7.37 | 4,475,616 |

## I  ADDITIONAL ABLATION STUDY

### I.1  CMT DUAL OR SINGLE-BRANCH

We modify the CMT's dual-branch Layer to a single-branch configuration and conduct additional experiments to assess the performance impact of integrating edge mesh and contact edge features. The biggest difference between edge mesh and contact mesh is that contact mesh has no relative displacement in its mesh space.

Table 12 shows the difference between the dual-branch and single-branch configurations within HCMT. It shows that better performance is achieved when using the dual-branch configuration.

Table 12: Ablation study results on whether to combine mesh and contact edges. The results of ablation studies (RMSE, rollout-all, $\times 10^3$).

| Model | Impact Plate | | Deformable Plate |
|---|---|---|---|
| | Position | Stress | Position |
| HMCT with CMT Dual-Branch | 20.34 | 14,447 | 7.67 |
| HMCT with CMT Single-Branch | 20.30 | 16,046 | 8.05 |

### I.2  PERFORMANCE IMPROVEMENT WITH REMESHING

Our proposed pooling method has a remeshing step. We perform additional ablation studies to evaluate the performance with and without remeshing. We proceed by setting the level of HCMT to 6. As shown in Table 13, we observe that, particularly for stress, the performance is better when the remeshing step is included.

Table 13: Ablation study on remeshing effect (RMSE, rollout-all, $\times 10^3$). A scaled Jacobian closer to 1 indicates a mesh with a shape closer to an equilateral triangle, signifying higher mesh quality.

| Model | Jacobian | Impact Plate | |
|---|---|---|---|
| | | Position | Stress |
| HCMT without remeshing | 0.32 | 19.21 | 17,482 |
| HCMT with remeshing | 0.51 | 20.34 | 14,447 |

### I.3  CLAMPING FOR NUMERICAL STABILITY

The process of clamping, i.e., the clip function in the self-attention of CMT and HMT, is used after taking the exponent of the internal term of the softmax function for numerical stability (Dwivedi & Bresson, 2020). We experiment to analyze the effect of clamping (see Equation 1, Equation 2, and Equation 5). Clamping ranges used for CMT and HMT are analyzed sequentially increasing from 1 to 4. We proceed by setting the level of HCMT to 6. Table 14 shows the position and stress RMSEs according to each clamping range. The optimal setting is from -2 to 2.

Table 14: Ablation study on numerical stability (RMSE, rollout-all, $\times 10^3$).

| Clamping range | Impact Plate | |
|---|---|---|
| | Position | Stress |
| -1 to 1 | 36.36 | 63.169 |
| -2 to 2 | 20.34 | 14,447 |
| -3 to 3 | 37.11 | 58.792 |
| -4 to 4 | 42.58 | 62.913 |

## J    SELF-ATTENTION MAPS

Figs 13, 14, 15, 16 show the self-attention maps of CMT's dual-branch and HMT.

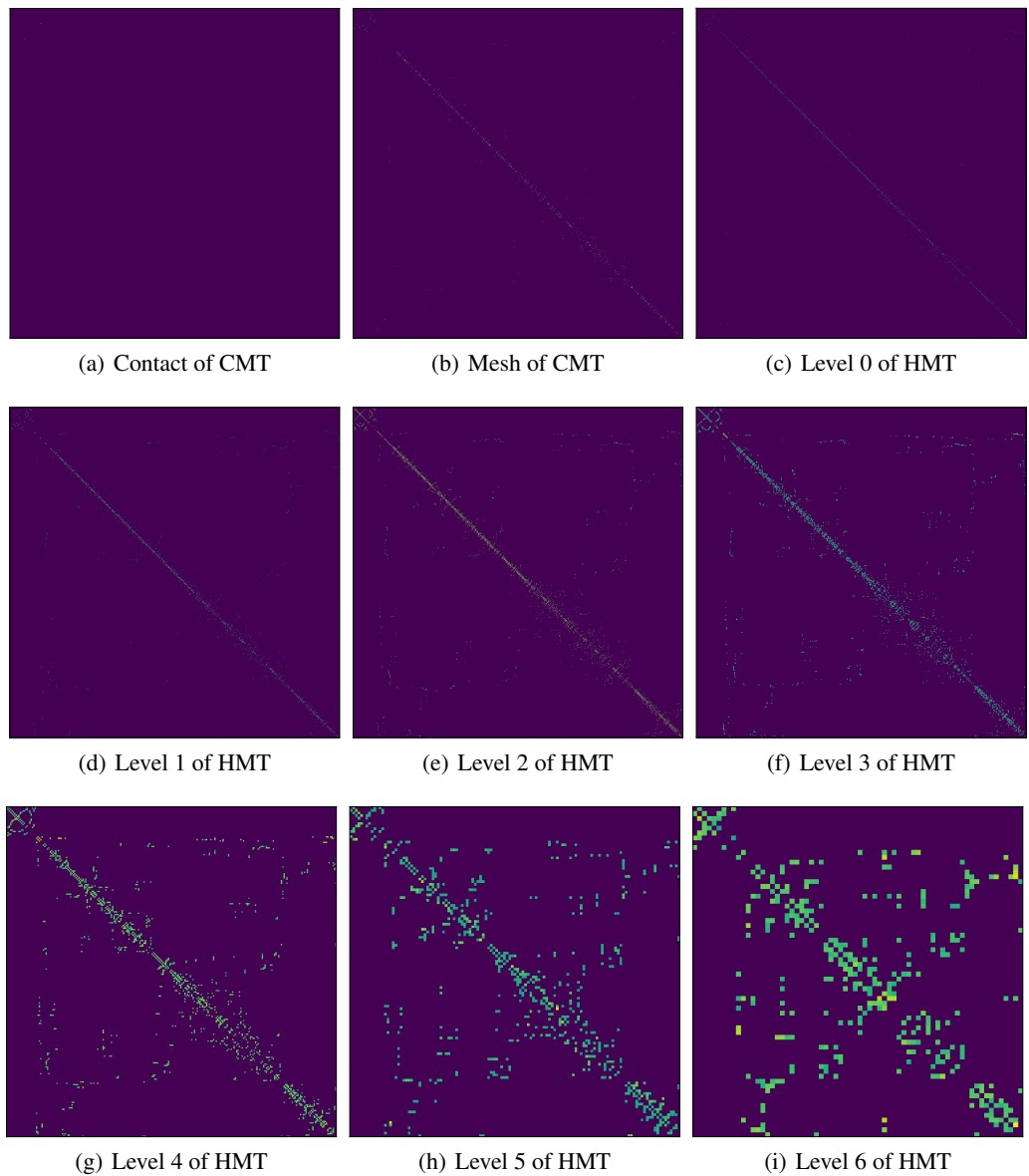

|                       |                       |                       |
|-----------------------|-----------------------|-----------------------|
| (a) Contact of CMT    | (b) Mesh of CMT       | (c) Level 0 of HMT    |
| (d) Level 1 of HMT    | (e) Level 2 of HMT    | (f) Level 3 of HMT    |
| (g) Level 4 of HMT    | (h) Level 5 of HMT    | (i) Level 6 of HMT    |

Figure 13: Self-attention maps of CMT and HMT on Impact Plate. In HMT, the top-left represents edges connected to plate nodes, while the bottom-right represents edges connected to ball nodes.

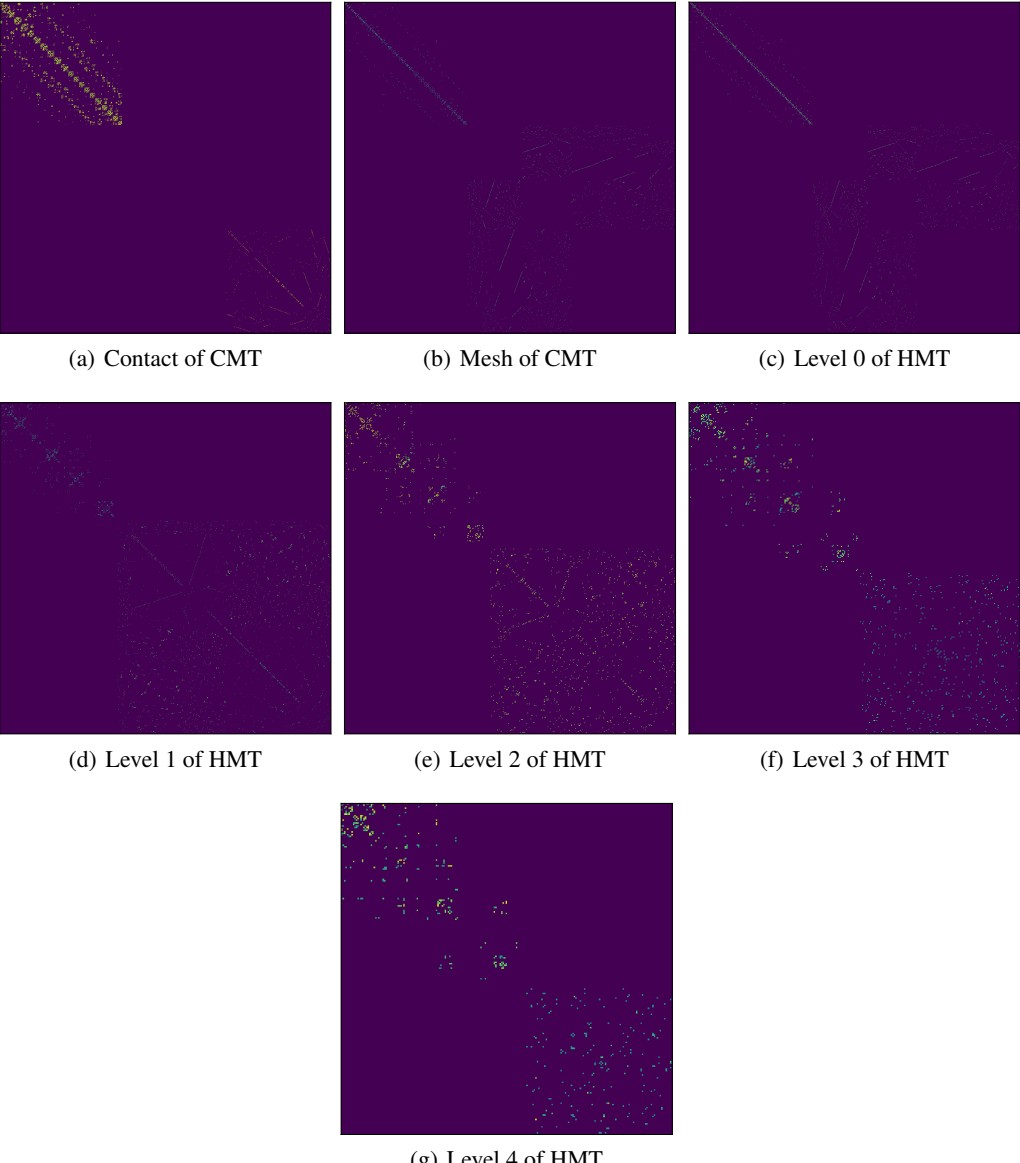

(a) Contact of CMT  (b) Mesh of CMT  (c) Level 0 of HMT

(d) Level 1 of HMT  (e) Level 2 of HMT  (f) Level 3 of HMT

(g) Level 4 of HMT

Figure 14: Self-attention maps of CMT and HMT on Sphere Simple. The reason it exhibits a different pattern compared to other domains is due to self-contacts. For example, when a cloth is folded, contacts occur among the cloth's nodes.

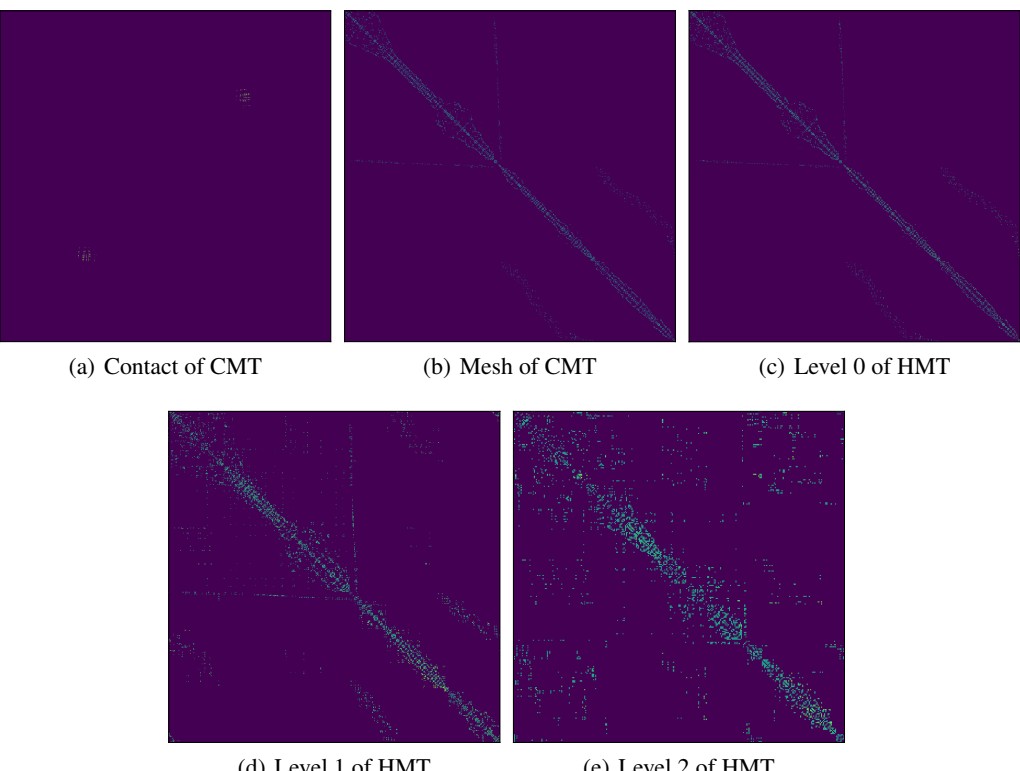

(a) Contact of CMT      (b) Mesh of CMT      (c) Level 0 of HMT

(d) Level 1 of HMT      (e) Level 2 of HMT

Figure 15: Self-attention maps of CMT and HMT on Deforming Plate. In the case of CMT, there is a slightly broader area of significance compared to Impact Plate because more of the obstacle's surface is involved in contact. Map of HMT shows an increasing number of meaningful connections as the level increases. The top-left represents connected mesh edges of the obstacle, while the bottom-right represents connected mesh edges of the plate

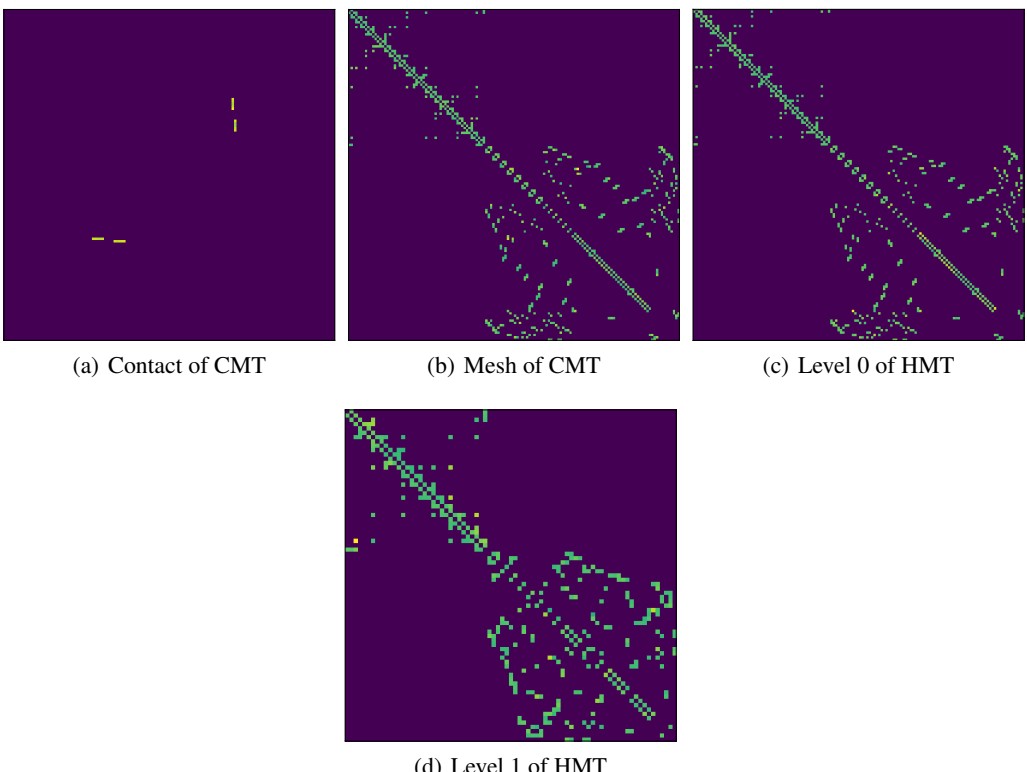

(a) Contact of CMT        (b) Mesh of CMT        (c) Level 0 of HMT

(d) Level 1 of HMT

Figure 16: Self-attention maps of CMT and HMT on Deformable Plate. Figure shows extensive feature maps in Deformable Plate, with more pronounced deformations in the plate than in the ball. The top-left represents edges connected to ball nodes, while the bottom-right represents edges connected to plate nodes.

# K    RESULT DETAILS

The results in each table are the rollout-all RMSE results for MGN, GT, and HCMT from five different seeds.

Table 15: MGN RMSE-all $\times 10^3$

| Seed | Deformable Plate | Sphere Simple | Deforming Plate | | Impact Plate | |
|------|------------------|---------------|----------|--------|----------|--------|
|      | Position | Position | Position | Stress | Position | Stress |
| 10 | 11.32 | 32.88 | 7.64 | 4,526,531 | 44.18 | 28,928 |
| 20 | 10.48 | 27.46 | 7.93 | 4,766,444 | 43.50 | 27,323 |
| 30 | 10.63 | 44.39 | 7.86 | 4,618,953 | 38.63 | 58,179 |
| 40 | 11.44 | 26.89 | 7.66 | 4,574,221 | 36.36 | 38,165 |
| 50 | 10.01 | 34.70 | 8.05 | 4,736,266 | 40.97 | 26,760 |
| mean | 10.78 | 33.26 | 7.83 | 4,644,483 | 40.73 | 35,871 |
| std | 0.54 | 6.33 | 0.16 | 92,520 | 2.94 | 11,893 |

Table 16: GT RMSE-all $\times 10^3$

| Seed | Deformable Plate | Sphere Simple | Deforming Plate | | Impact Plate | |
|------|------------------|---------------|----------|--------|----------|--------|
|      | Position | Position | Position | Stress | Position | Stress |
| 10 | 14.13 | 371.79 | 11.21 | 9,014,384 | 61.05 | 26,804 |
| 20 | 13.05 | 189.89 | 11.44 | 9,332,153 | 60.23 | 25,328 |
| 30 | 14.31 | 134.32 | 11.83 | 9,403,657 | 64.97 | 28,948 |
| 40 | 13.82 | 77.75 | 11.18 | 9,043,427 | 51.44 | 33,372 |
| 50 | 13.38 | 445.48 | 11.04 | 9,047,868 | 58.22 | 82,005 |
| mean | 13.74 | 243.85 | 11.34 | 9,168,298 | 59.18 | 39,291 |
| std | 0.47 | 141.08 | 0.28 | 164,941 | 4.45 | 21,529 |

Table 17: HCMT RMSE-all $\times 10^3$

| Seed | Deformable Plate | Sphere Simple | Deforming Plate | | Impact Plate | |
|------|------------------|---------------|----------|--------|----------|--------|
|      | Position | Position | Position | Stress | Position | Stress |
| 10 | 7.67 | 28.30 | 7.37 | 4,475,616 | 20.34 | 14,447 |
| 20 | 8.05 | 30.95 | 7.56 | 4,609,359 | 21.70 | 15,685 |
| 30 | 7.35 | 30.06 | 7.53 | 4,489,145 | 20.92 | 14,257 |
| 40 | 8.18 | 29.40 | 7.56 | 4,534,407 | 20.11 | 14,524 |
| 50 | 7.07 | 33.35 | 7.44 | 4,571,253 | 20.45 | 14,798 |
| mean | 7.67 | 30.41 | 7.49 | 4,535,956 | 20.71 | 14,742 |
| std | 0.42 | 1.71 | 0.07 | 49,937 | 0.57 | 502 |

Table 18: Detailed measurements

| Metrics | Datasets | | GT | MGN | HCMT |
|---|---|---|---|---|---|
| RMSE-1 $\times 10^3$ | Deformable Plate | Position | $0.886_{\pm 0.006}$ | $0.779_{\pm 0.011}$ | $0.724_{\pm 0.006}$ |
| | Sphere Simple | Position | $0.071_{\pm 0.006}$ | $0.076_{\pm 0.004}$ | $0.121_{\pm 0.015}$ |
| | Deforming Plate | Position Stress | $0.178_{\pm 0.009}$ $2,919,024_{\pm 154,061}$ | $0.129_{\pm 0.007}$ $1,461,039_{\pm 332,815}$ | $0.128_{\pm 0.008}$ $1,161,493_{\pm 87,110}$ |
| | Impact Plate | Position Stress | $0.031_{\pm 0.002}$ $676_{\pm 8}$ | $0.029_{\pm 0.002}$ $686_{\pm 12}$ | $0.023_{\pm 0.001}$ $682_{\pm 13}$ |
| RMSE-all $\times 10^3$ | Deformable Plate | Position | $13.74_{\pm 0.47}$ | $10.78_{\pm 0.54}$ | $7.67_{\pm 0.42}$ |
| | Sphere Simple | Position | $243.85_{\pm 141.08}$ | $33.26_{\pm 6.33}$ | $30.41_{\pm 1.71}$ |
| | Deforming Plate | Position Stress | $11.34_{\pm 0.28}$ $9,168,298_{\pm 164,941}$ | $7.83_{\pm 0.16}$ $464,483_{\pm 92,520}$ | $7.49_{\pm 0.07}$ $4,535,956_{\pm 49,937}$ |
| | Impact Plate | Position Stress | $59.18_{\pm 4.45}$ $39,291_{\pm 21,529}$ | $40.73_{\pm 2.94}$ $35,871_{\pm 11,893}$ | $20.71_{\pm 0.57}$ $14,742_{\pm 502}$ |
| Training time/step (ms) | Deformable Plate | | 55.65 | 38.63 | 53.16 |
| | Sphere Simple | | 130.36 | 89.28 | 59.02 |
| | Deforming Plate | | 76.93 | 51.11 | 53.53 |
| | Impact Plate | | 79.31 | 51.56 | 51.12 |

## L    OTHER VARIABLE CONTOUR AND ROLLOUT IMAGES

Figs. 17, 18, 19, 20 are rollout images of Impact Plate, Deforming Plate, Sphere Simple, and Deformable Plate, over time or step flow.

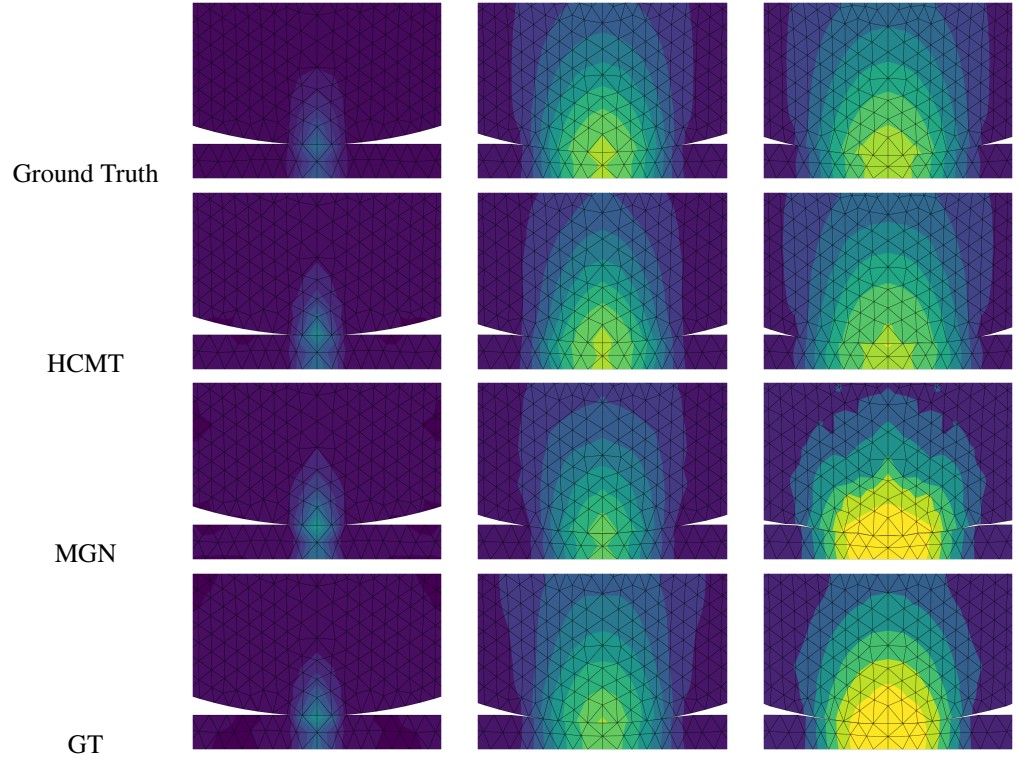

Figure 17: The stress contours of various model-predicted rollouts compared to the ground truth at Impact Plate.

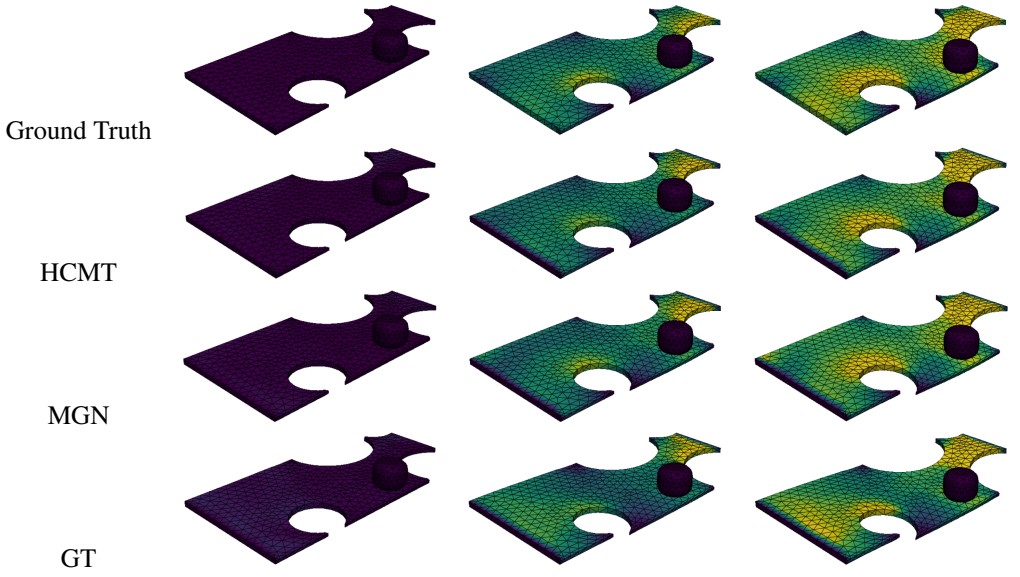

Figure 18: The stress contours of various model-predicted rollouts compared to the ground truth at Deforming Plate.

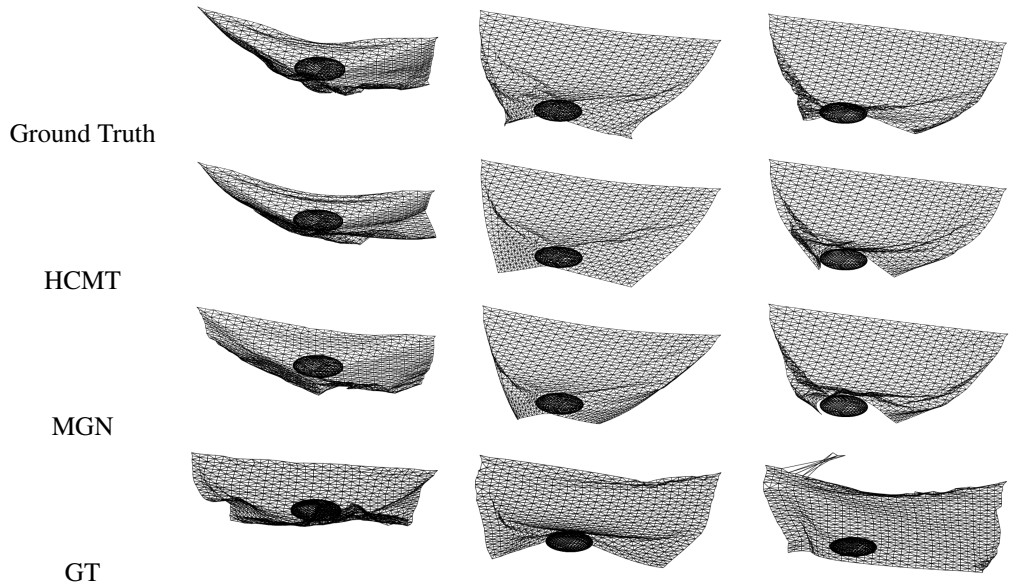

Figure 19: The image of various model-predicted rollouts compared to the ground truth at Sphere Simple.

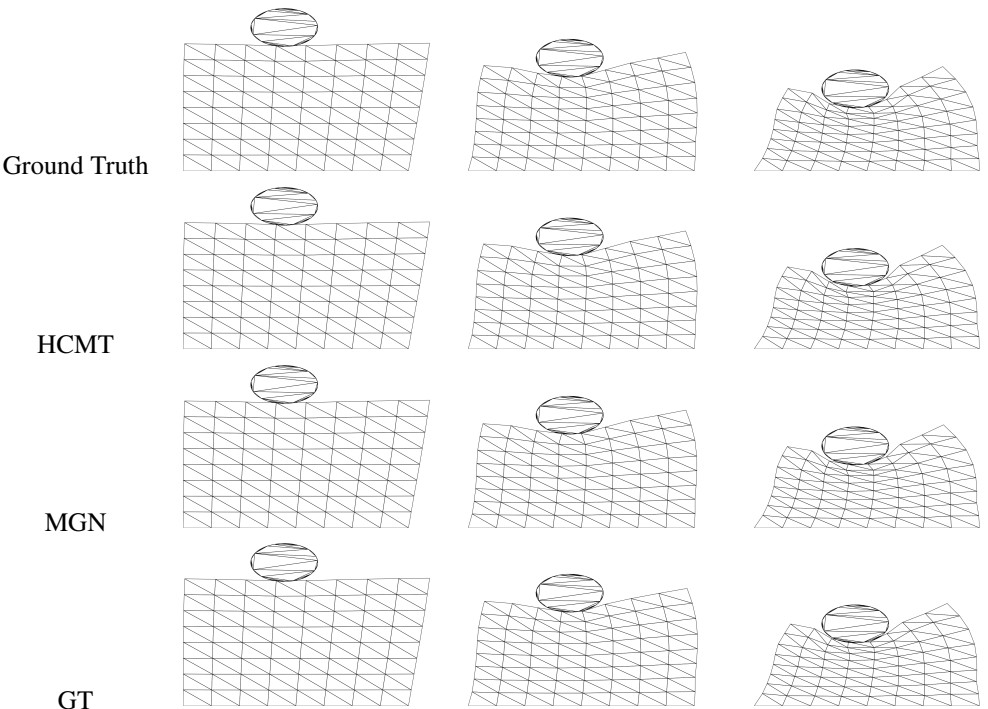

Figure 20: The image of various model-predicted rollouts compared to the ground truth at Deformable Plate.

