# OpenReview forum: "Learning Flexible Body Collision Dynamics with Hierarchical Contact Mesh Transformer"
_ICLR.cc/2024/Conference — ICLR 2024 poster_

### Official Review · Reviewer_gSaT · 2023-10-30

**Soundness:** 3 good
**Presentation:** 3 good
**Contribution:** 3 good
**Rating:** 6
**Confidence:** 4

**Summary:**

In this paper, a novel dual graph transformer architecture is proposed for flexible body collisions. The underlying architecture is built by combining a traditional Collision Mesh Transformer (CMT) with a novel hierarchical dual-branch attention Hierarchical Mesh Transformer (HMT) to enable long-distance interactions that are important for collision dynamics. The paper evaluates their contributions on four datasets (representing different classes of interaction dynamics), showing how the proposed method outperforms existing Graph Transformer (GT) and MeshGraphNet (MGN) architectures in all cases. The authors furthermore perform some ablation studies for some aspects of their methodology on the same datasets. Finally, the authors conclude by pointing out some limitations addressable in future work, e.g., including learnable pooling methods.

**Strengths:**

The originality of the paper in its application domain, i.e., regarding flexible collision dynamics, seems strong as this area of physical simulations is not really explored so far using Neural Networks. However, the originality of the proposed method itself, i.e., the use of two graph transformers, is not as strong.

I believe the main strengths of the contribution of the paper are demonstrated by Tables 2 and 9 as well as Figure 7. The results show a clear benefit to using the proposed transformer-based architecture on the chosen datasets. Furthermore, the authors provide a new dataset useful for future research.

The writing of the overall paper is good, and the explanations are mostly straightforward to understand from a linguistic standpoint, but not several questions arise when reading the paper.

Regarding Significance: On one hand, the authors state that the proposed network could be useful in product design (as stated in the abstract and conclusions), which is seemingly a positive aspect, but this is difficult to verify from the submission. On the other hand, the usage of two separate graph transformers with different architectures, i.e., non-hierarchical and hierarchical, seems an interesting point that could be explored in other applications areas as well.

**Weaknesses:**

The overall approach of the paper seems technically sound, but there are several points in the paper that raise questions that should be addressed. Furthermore, the core contribution of the paper is not really clear enough, as it is not clear how novel the use of the double transformer architecture is. The authors point out that several hierarchical approaches already exist in the literature (in the related work section), but it is not clear how different they are from the proposed architecture. Furthermore, the datasets are not easy to understand just by looking at the provided information, i.e., what is actually happening in the impact plate scenario? The following is a list of concerns, weak points and linguistic problems in the paper that the authors could take to heart and clarify if possible (not listed in order of severity):
- In the abstract, the authors mention that their proposed 'unique flexible body dynamics dataset'  is commonly used for product design. It is not clear how such a dataset is useful for product design, and the use of the word 'commonly' combined with a 'unique dataset' seems contradictory. The authors should clarify how this dataset is useful.
- The authors claim that GNNs are capable of 'intelligently distributing computation to areas of interest', but this seems somewhat dubious as a claim without any reference to support the argument.
- The authors claim that classic GNNs 'cannot quickly propagate the influence of collisions over long distances' and justify this by a requirement of needing multiple GNN steps to propagate information. However, hierarchical mesh transformers, as pointed out by the authors in the related work section, are clearly capable of addressing such issues already. Furthermore, the proposed architecture also requires multiple propagation steps.
- It is unclear how Figure 2 is different from existing hierarchical transformers
- What is 'N' when describing the computational complexity of Transformers?
- In the related work section, the authors describe that GNNs are 'expected to be applicable to all systems', but this is clearly not the case, as the authors already demonstrate in their results. This should be clarified.
- The authors state that slow propagation leads to unnatural dynamics, but this does not seem obvious. If propagation is slow, couldn't one add more layers until the same receptive fields and propagation distances are achieved?
- What is a "contact edge", and how are they computed? The explanations in Section 3.2 are very sparse, and this seems to be a central point of the paper. Also, do they include self-interactions or not?
- The authors mention 'the concatenated features mentioned earlier', but the discussion on features is moved to the appendix. This should be streamlined.
- What is the meaning of the double hat notation in (3)? This is made more confusing by the authors using the hat for a pre-layer norm and later using a dot + hat combination in the decoder and updater paragraph.
- In (1) and (2), the authors should make use of \left( and \right) in latex.
- The authors mention that in 'fluid dynamics, node, and edge lengths are always fixed', which is not true in Lagrangian simulations, e.g., in the CConv approach by Ummenhofer et al.
- The authors note that Figure 6 'shows that as the level increases, shortcut message passing is possible in our method', but this is not really clear from the figure, e.g., is this related to intra or inter-body messages or both?
- The self-attention maps shown in the main paper and appendix do not provide many insights into the learned behavior and are not very intuitive to understand.
- Figure 7 should include colormaps.
- The use of a scaling factor for the results in Table 2 is not ideal and very confusing to read right next to unscaled results in Figure 9.

**Questions:**

- Most importantly, how different is the proposed HCMT architecture from existing hierarchical models? Is the novelty just in using contact edges, using a transformer, or something else?
- From the descriptions, it is unclear how contact edges are computed, i.e., are they based on a radius search? And if so, what search radii are used? What influence does this have on the achieved results, and would using a non-hierarchical transformer with a similar search radius perform well? Also, how large is the used radius in relation to the mesh elements? The appendix makes some mention of a parameter gamma (Table 8), but this parameter seems to vary significantly between datasets, but there is no clear argument as to why this is the case either.
- A significant concern with the dataset is the apparent exclusion of two trajectories where stress singularities occur. These seem to exist in the solver used for the data generation, so shouldn't a neural network that learns a simulator's behavior be expected to also lead to the same results? Picking and Choosing from datasets is not ideal, and the authors should include a comparison of the different methods with and without these singularities. If the proposed method works best with and without these singularities, then they should be included; if the proposed method works worse than existing methods if they are included, then this should be discussed more straightforwardly.

- Figure 2 and Figure 4 seem to conflict with each other. Figure 2 shows how the interactions appear to be computed simultaneously for different distances, whereas Figure 4 shows how the different levels of the hierarchy are computed one after another. Which one is used and why?
- The authors mention that they 'clamp' values for numerical stability, which seems important but is not further discussed. Similarly, the authors mention that they use a pre-layer norm, denoted by a hat, but (4) uses another layer norm after applying the transformer?
- Are the pooling operations performed globally or only per body? The figures seem to indicate the latter, but the text is unclear.
- Table 3 raises some concerns regarding the reproducibility of the results. First of all, it is unclear how many seeds are used for all evaluations and how the standard deviations are computed, e.g., are they computed within an evaluation dataset or across different seeds? Furthermore, for example, Table 2 shows a position and stress RMSE for the deforming plate of 7.49 pm 0.07 and 4,535,956 pm 49,937, respectively. Whereas Table 3 shows a position RMSE of 7.37, which is 0.12 away from Table 2, and a stress RMSE of 4,475,616, which is 60,340 away from Table 3. Both results, thus, are outside of the given standard deviation. Why is this the case?
- Why does the error increase significantly with more levels for the deformable plate scenario? The authors note that a decrease in levels decreases the RMSE, but this does not really seem sufficient as not having any levels would mean that there is no need for long-range interactions, which should also be possible to do using MGN, but MGN performs worse than even the longest-range HCMT network.

============

I want to thank the authors for the detailed information and the updates. Overall, I'm still leaning towards the positive side. I would like to slightly update my score, but given the discrete steps I will stick to the current one and let the AC know about my "slight raise".

---

> ### Author Response · Authors · 2023-11-17
>
> We appreciate Reviewer gSaT for the positive feedback and insightful comment, highlighting our strengths in
> 1. New research using neural networks in the flexible collision dynamics
> 2. Demonstrates the benefits of using a Transformer-based architecture on the main table
> 3. Provide a new valuable dataset for future research
> 4. Clear writing
>
> Below, we carefully address your concerns. We uploaded the revised paper, where changes are highlighted in red.
>
> ---
>
> **Q1. In the abstract, the authors mention that their proposed 'unique flexible body dynamics dataset' is commonly used for product design. It is not clear how such a dataset is useful for product design, and the use of the word 'commonly' combined with a 'unique dataset' seems contradictory. The authors should clarify how this dataset is useful.**
>
> We agree that it may appear contradictory. The dataset mentioned in the abstract is used to represent flexible body dynamics typically encountered in product design scenarios in the electronic display industry. Therefore, we revised the abstract to reflect this as follows.
>
> *"Lastly, we propose a flexible body dynamics dataset consisting of trajectories that reflect experimental settings frequently used in the display industry for product designs."*
>
> Also, we provide additional descriptions of the dataset as follows:
> The evaluation of Impact Plate follows industry-standard practices, primarily assessing the stiffness of panels [1]. We considered the dataset to optimize the design of the plate (e.g., display panel) as a downstream task. The plate thickness and material properties related to product design, as well as the ball size, mesh size, and drop height corresponding to the evaluation conditions were set as parameters. This is because the rigidity of the plate varies depending on the material characteristics and thickness of the plate, and the evaluation method also requires different performance values. If the stress value occurring inside the plate exceeds the ultimate strength, damage occurs, so the material and thickness are selected so that the strength does not exceed. Therefore, checking the stress value can be seen as a more important factor for product design. In addition, because the analysis speed is faster than the existing ANSYS Simulator, design optimization can be reached through more experiments. Related content has been added and modified in Appendix C.1 and marked in red.
>
> > [1] Liang Xue et al. "Dynamic analysis of thin glass under ball drop impact with new metrics." International Electronic Packaging Technical Conference and Exhibition. Vol. 55751. American Society of Mechanical Engineers, 2013.
>
> **Q2. The authors claim that GNNs are capable of 'intelligently distributing computation to areas of interest', but this seems somewhat dubious as a claim without any reference to support the argument.**
>
> Our intention was that it could be applied to a variety of systems of interest, but it was misleading and the wording was unclear, so we revised it.
>
>
> **Q3. The authors claim that classic GNNs 'cannot quickly propagate the influence of collisions over long distances' and justify this by a requirement of needing multiple GNN steps to propagate information. However, hierarchical mesh transformers, as pointed out by the authors in the related work section, are clearly capable of addressing such issues already. Furthermore, the proposed architecture also requires multiple propagation steps.**
>
> We would like to clarify that given the same number of propagation steps, the proposed HCMT model can propagate further compared to classic GNNs/dual-level or multi-level GNNs/previous hierarchical transformers (indeed, there are only dual-level transformers). We would like to refer you to the discussion in [Q7](https://openreview.net/forum?id=90yw2uM6J5&noteId=Qxx62uwYeu) and [Q17](https://openreview.net/forum?id=90yw2uM6J5&noteId=3Mhra69reC) for more details.
>
>
> **Q4. It is unclear how Figure 2 is different from existing hierarchical transformers**
>
> Figure 2 is a conceptual illustration only for showing how collision propagation occurs at each level of the HMT. Mesh coarsening, which happens at each level of HMT, cannot be depicted in a single static figure. This is to demonstrate the necessity of long-distance propagations in the flexible dynamics where contacts occur.
>
> Please check our response to [Q17](https://openreview.net/forum?id=90yw2uM6J5&noteId=3Mhra69reC) for differences from existing hierarchical models.
>
>
> **Q5. What is '$N$' when describing the computational complexity of Transformers?**
>
> Thank you for pointing out. $N$ refers to the number of nodes, and the related content has been modified and reflected in red in Section 1.

---

> ### Author Response · Authors · 2023-11-17
>
> **Q6. In the related work section, the authors describe that GNNs are 'expected to be applicable to all systems', but this is clearly not the case, as the authors already demonstrate in their results. This should be clarified.**
>
> We admitted that our expression was incorrect, and it was corrected and marked in red.
>
> **Q7. The authors state that slow propagation leads to unnatural dynamics, but this does not seem obvious. If propagation is slow, couldn't one add more layers until the same receptive fields and propagation distances are achieved?**
>
> Ambiguous sentences have been corrected for clarity and marked in red. Please see our revised Section 2.1.
>
> Here is the summary of our intention. As pointed out, MGN can add more layers to achieve larger receptive fields and propagation distances. However, considering computational efficiency, the number of layers can be fixed to a certain "reasonable" number (i.e., a hyperparameter) and certain choices of this number could result in inappropriate modeling results due to the small receptive field.
>
> However, given the same number of layers, the receptive field of the proposed model becomes much larger and the issue caused by having the small receptive field can be resolved. The following table shows that the number of receptive fields increases linearly in the case of MGN, while it increases exponentially in HCMT due to the multi-level mesh coarsening
>
> | Number of layers | MGN | HCMT |
> |:----------------:|:---:|:----:|
> |         1        |  1  |   1  |
> |         2        |  2  |   2  |
> |         3        |  3  |   4  |
> |         4        |  4  |   8  |
> |         5        |  5  |  16  |
> |         6        |  6  |  32  |
>
>
> **Q8. What is a "contact edge", and how are they computed? The explanations in Section 3.2 are very sparse, and this seems to be a central point of the paper. Also, do they include self-interactions or not?**
>
> Thank you for your feedback. Here's the detailed description.
>
> When converting from $M^t$ to $G_0$, the contact edges are computed. A contact edge is a connection between two different objects' nodes or a self contact within an object. For every node, we find all neighboring nodes within a radius and construct them as contact edges excluding previously connected mesh edges. Section 3.2 was written to make it easier to understand from the perspective of the overall flow. An explanation of contact edges and calculation methods has been added to “Features and Encoder Layer” in Section 3.3.
>
> We also provide the information stating that except for Sphere Simple, self-contacts do not occur. Relevant information can be found in Table 6 in Appendix C.
>
>
> **Q9. The authors mention 'the concatenated features mentioned earlier', but the discussion on features is moved to the appendix. This should be streamlined.**
>
>
> Thanks for pointing out. This definitely is a mistake and we have modified the relevant part in Section 3.3 and marked it in red.
>
>
> **Q10. What is the meaning of the double hat notation in (3)? This is made more confusing by the authors using the hat for a pre-layer norm and later using a dot + hat combination in the decoder and updater paragraph**
>
> Thanks for pointing out. We updated $\hat{\hat{\mathbf{z}}}_i$ to $\bar{\mathbf{z}}_i$.
>
>
> **Q11. In (1) and (2), the authors should make use of \left( and \right) in latex.**
>
> Thank you for the suggestion. Eq. (1) and (2) have been modified and marked in red.
>
>
> **Q12. The authors mention that in 'fluid dynamics, node, and edge lengths are always fixed', which is not true in Lagrangian simulations, e.g., in the CConv approach by Ummenhofer et al.**
>
> The reviewer is correct. We limited the discussion to “Eulerian”. Please refer to the new text in Section 3.3.
>
>
> **Q13. The authors note that Figure 6 'shows that as the level increases, shortcut message passing is possible in our method', but this is not really clear from the figure, e.g., is this related to intra or inter-body messages or both?**
>
> Figure 6 represents “inter-body messages” and occur in HMT. We added corresponding texts to the caption of Figure 6 and marked it in red.

---

> ### Author Response · Authors · 2023-11-17
>
> **Q14. The self-attention maps shown in the main paper and appendix do not provide many insights into the learned behavior and are not very intuitive to understand.**
>
> Here we provide our interpretation on the attention maps.
>
> Figure 8(a) shows a colormap of the contact attention distribution in CMT, showing different levels of importance. Brighter dots indicate connections between center nodes on the surface of the ball and the plate. They show that contact propagation occurring at the center is significant.
>
> Figure 8(b) shows the mesh attention distribution for each object in CMT. The x and y axes are both node indices, and the parts represented by blue boxes represent connections among plate nodes. If the points are distributed far along the diagonal, it means that there are interactions among distant nodes.
>
> Figure 13 in Appendix H shows the self-attention map of the CMT or HMT relative to Impact Plate. As the level of HMT increases, more points farther away from the diagonal appear, indicating interactions among more distant nodes.
>
>
> **Q15. Figure 7 should include colormaps.**
>
> Colormaps were added to Figure 7.
>
> **Q16. The use of a scaling factor for the results in Table 2 is not ideal and very confusing to read right next to unscaled results in Figure 9.**
>
> Following the suggestion, we update numbers accordingly.
>
>
> **Q17. Most importantly, how different is the proposed HCMT architecture from existing hierarchical models? Is the novelty just in using contact edges, using a transformer, or something else?**
>
> Here we would like to clarify the differences from other methods and emphasize the contributions we made.
>
> HCMT is designed to efficiently deliver impact through contacts. Recent mesh-based hierarchical models use dual-level approaches [1-3], but our model has multiple levels (see Table below). The biggest advantage of multi-level is that as the number of levels increases, it can propagate exponentially further. In particular, we used the Delaunay triangulation to improve the mesh quality in the pooling method. Additionally, using two Transformers has important implications. As shown in Table 3 of the paper, using HMT or CMT alone does not produce good performance. This indicates that contact propagations can be efficiently processed by first using the CMT layer and then by using the HMT layer, which propagates a contact over long distances.
>
> Despite the many and important contact situations (e.g., automobile impact testing) in many industries, recent mesh-based GNN papers focus on fluid dynamics and overlook structural mechanics. Structural mechanics problems based on Lagrangian tend to have challenges in different perspectives because they show strong nonlinear characteristics due to contacts, which needs to be addressed properly. Our model focuses on the structural mechanics problems and addresses those challenges via novel components that are proposed in this paper.
>
> |          Model                               | Hierarchical architecture |
> | -------------------------------------------- |---------------------------|
> |        MS-GNN[1]        |       Dual-level GNN      |
> |       BSMS-GNN[4]                            |      Multi-level GNN      |
> | GMR-Transformer-GMUS[2], EAGLE Transformer[3]                    |   Dual-level Mesh Transformer  |
> |           HCMT                               |  Multi-level Mesh Transformer  |
>
>
> > [1] Meire Fortunato et al. "MultiScale MeshGraphNets." ICML 2022 2nd AI for Science Workshop, 2022.
> >
> > [2] Xu Han et al. "Predicting Physics in Mesh-reduced Space with Temporal Attention." ICML, 2022
> >
> > [4] Steeven Janny et al. "EAGLE: Large-Scale Learning of Turbulent Fluid Dynamics with Mesh Transformers." ICLR, 2023.
> >
> > [5] Yadi Cao et al. "Bi-Stride Multi-Scale Graph Neural Network for Mesh-Based Physical Simulation." ICML, 2023

---

> ### Author Response · Authors · 2023-11-17
>
> **Q18. From the descriptions, it is unclear how contact edges are computed, i.e., are they based on a radius search? And if so, what search radii are used? What influence does this have on the achieved results, and would using a non-hierarchical transformer with a similar search radius perform well? Also, how large is the used radius in relation to the mesh elements? The appendix makes some mention of a parameter gamma (Table 8), but this parameter seems to vary significantly between datasets, but there is no clear argument as to why this is the case either.**
>
> When converting from $M^t$ to $G_0$, the contact edges are computed. For each node, we find all neighboring nodes within a radius and configure them as contact edges (($|\mathbf{x}_i- \mathbf{x}_q| < \gamma$)), excluding previously connected mesh edges. The relevant information is mentioned in Appendix C. According to MGN, the radius is an important hyperparameter and is different for each dataset.
>
> According to MGN[1] and GGNS[3], radius is an important hyperparameter and was tuned. We similarly performed radius tuning for Impact Plate using the MGN model and selected the optimal radius (0.4mm). HCMT used this selected optimal radius without tuning. For the remaining datasets, the values mentioned in the corresponding papers[1, 2] were used.
>
> The table shows RMSE results for each radius with the MGN model. The distribution of the Impact Plate mesh size is 0.3-0.5mm, and the optimal results were obtained when the radius was set to 0.4mm, and the optimal results were obtained when the mesh size and radius are similar.
>
> Our best guess is that the reason there are differences in the radius for datasets is because the mesh sizes used when creating them are all different. (Again, Impact Plate is the only dataset that we have direct access to.) If the mesh size is small and the collision recognition radius is wide, more edge sets are formed, which increases computing resources. If the mesh size is large and the collision recognition radius is small, contact may not be recognized well.
>
> | Radius (mm) | Pos. RMSE |   Stress RMSE  |
> |:-----------:|:---------:|:--------------:|
> |     0.1     |   51.37   |        32,314  |
> |     0.2     |   55.64   |        49,221  |
> |     0.3     |   52.89   |        26,022  |
> |     0.4     |   44.18   |        28,928  |
> |     0.5     |   57.42   |        27,065  |
>
> > [1] Tobias Pfaff et al. "Learning Mesh-Based Simulation with Graph Networks." ICLR, 2021.
> >
> > [2] Jonas Linkerhägner et al. "Grounding Graph Network Simulators using Physical Sensor Observations." ICLR 2023 Workshop on Physics for Machine Learning, 2023.
> >
> > [3] A. Sanchez-Gonzalez et al. "Learning to Simulate Complex Physics with Graph Networks." ICML, 2020.
>
>
> **Q19. A significant concern with the dataset is the apparent exclusion of two trajectories where stress singularities occur. These seem to exist in the solver used for the data generation, so shouldn't a neural network that learns a simulator's behavior be expected to also lead to the same results? Picking and Choosing from datasets is not ideal, and the authors should include a comparison of the different methods with and without these singularities. If the proposed method works best with and without these singularities, then they should be included; if the proposed method works worse than existing methods if they are included, then this should be discussed more straightforwardly.**
>
> The reviewer is correct. The table below shows the results of experiments with and without stress singularities. When including stress singularities, the position and stress RMSEs increase for MGN and our method. Nonetheless, HCMT shows lower errors than MGN. Experimental results have been added to Appendix L.
>
> We found that training losses sometimes fluctuate with stress singularities. The average maximum stress for all trajectories is 700,464, but for 173 and 233 trajectories, they are 434,718,530 and 43,678,880, respectively.
>
> Moreover, we overthought from aour domain perspective. If a stress singularity occurs during a simulation using a numerical analyzer (e.g., ANSYS and COMSOL), simulation engineers do not use it because singularities do not occur in the real world.
>
> | Model | Pos. w/Singularity | Pos. w/o Singularity | Stress w/Singularity | Stress w/o Singularity |
> |:-----:|:------------------:|:--------------------:|:--------------------:|:----------------------:|
> |  MGN  |      7.98          |        7.64          |    5,085,048         |        4,526,531       |
> |  HCMT |      7.77          |        7.37          |    4,995,446         |        4,475,616       |

---

> ### Author Response · Authors · 2023-11-17
>
> **Q20. Figure 2 and Figure 4 seem to conflict with each other. Figure 2 shows how the interactions appear to be computed simultaneously for different distances, whereas Figure 4 shows how the different levels of the hierarchy are computed one after another. Which one is used and why?**
>
> The reviewer's interpretation on Figure 4 is correct. We again emphasize that Figure 2 is a concept illustration to make it easier to understand. To avoid this, we update the caption in Figure 2 and marked in red.
>
>
> **Q21. The authors mention that they 'clamp' values for numerical stability, which seems important but is not further discussed. Similarly, the authors mention that they use a pre-layer norm, denoted by a hat, but (4) uses another layer norm after applying the transformer?**
>
> We agree with the reviewer on the point that the clamp function is important. We selected the optimal range through our hyperparameter tuning. The results were added to Appendix G.3 and also mentioned in Section 3.3. The following table shows the results of evaluation on Impact Plate with the HCMT model. Optimal results were obtained when the clamping range was set to between -2 and 2.
>
> | Clamping range | Position RMSE | Stress RMSE |
> |:-----:|:-----------:|:-----------:|
> |   -1 to 1   |   36.36   |    63,169   |
> |   -2 to 2   |   20.34   |    14,447   |
> |   -3 to 3   |   37.11   |    58,792   |
> |   -4 to 4   |   42.58   |    62,913   |
>
> **Q22. Are the pooling operations performed globally or only per body? The figures seem to indicate the latter, but the text is unclear.**
>
> We would like to clarify that the pooling method is applied only per body. The relevant information has been added to the Pooling Method syntax in Section 3.3 and marked in red.
>
>
> **Q23. Table 3 raises some concerns regarding the reproducibility of the results. First of all, it is unclear how many seeds are used for all evaluations and how the standard deviations are computed, e.g., are they computed within an evaluation dataset or across different seeds? Furthermore, for example, Table 2 shows a position and stress RMSE for the deforming plate of 7.49 pm 0.07 and 4,535,956 pm 49,937, respectively. Whereas Table 3 shows a position RMSE of 7.37, which is 0.12 away from Table 2, and a stress RMSE of 4,475,616, which is 60,340 away from Table 3. Both results, thus, are outside of the given standard deviation. Why is this the case?**
>
> We use 5 different seeds and calculate the mean and standard deviation based on the results evaluated on test data. The standard deviation is calculated relative to the entire population given as an argument. All calculated values are calculated to 14 decimal places and are rounded to 2 decimal places in the table. Results for each model for all datasets can be found in Tables 11, 12, and 13 in Appendix I.
>
> The position and stress RMSEs are not outliers because Z-scores for position RMSE and stress RMSE are -1.71 and -1.20, respectively. The confidence intervals for position and stress are 95% and 88%, respectively.
>
> **Q24. Why does the error increase significantly with more levels for the deformable plate scenario? The authors note that a decrease in levels decreases the RMSE, but this does not really seem sufficient as not having any levels would mean that there is no need for long-range interactions, which should also be possible to do using MGN, but MGN performs worse than even the longest-range HCMT network.**
>
> Our conjecture for the reason errors increase as the level increases is that Deformable Plate has more than 10 times fewer nodes than other datasets. Table 4 shows the number of nodes according to the level of the Deformable Plate dataset. The number of nodes in $G_0$ is 138, and the number of nodes decreases every step the level goes up one level. It is believed to be a dataset that can already propagate contacts with just one pooling. If the experiment is conducted with an increased number of nodes by reducing the mesh size used for Deforable Plate in order to improve the accuracy of simulation, it would be advantageous to have higher levels.
>
> |     Datasets     | Level 0 | Level 1 | Level 2 | Level 3 | Level 4 | Level 5 | Level 6 |
> |:----------------:|:-------:|:-------:|:-------:|:-------:|:-------:|:-------:|:-------:|
> | Deformable Plate |   138   |    77   |    48   |    32   |    24   |    20   |    18   |
> |   Impact Plate   |   2208  |   1108  |   559   |   285   |   148   |    78   |    42   |
> |   Sphere Simple  |   1731  |   898   |   523   |   330   |   224   |   170   |   138   |
> |  Deforming Plate |   1276  |   658   |   344   |   183   |    98   |    55   |    32   |

---

> ### Author Response · Authors · 2023-11-22
>
> We appreciate the reviewer’s time and effort in reviewing our manuscript and insightful comments. As the closure of the discussion period is approaching, we would like to bring the review’s attention and check if the reviewer could let us know whether the concerns or the misunderstanding have been addressed. If this is the case, we would appreciate if you could adjust your rating accordingly. Thank you!

---

### Official Review · Reviewer_zh4D · 2023-10-31

**Soundness:** 2 fair
**Presentation:** 3 good
**Contribution:** 3 good
**Rating:** 6
**Confidence:** 2

**Summary:**

This paper presents a transformer-based neural network model for learning collision/contact responses in flexible dynamics. The network consists of two transformers capturing the mesh and contact dynamics, respectively. The results are evaluated against standard neural network baselines including MeshGraphNet.

**Strengths:**

I’ll admit first that I am not an expert in “flexible dynamics”. Although I recognize the governing equations in Table 1, the concept of “flexible dynamics,” which seems to be rooted in the engineering community, is quite new to me. Overall, I am on the fence but with very low confidence due to this reason.

At a high level, I think capturing the long-distance collision response with a hierarchical network architecture is an interesting and reasonable idea. While hierarchical networks have been applied in other physical/numerical systems as reviewed in the related work, using them to capture contact/collision seems novel to me.

**Weaknesses:**

I will briefly summarize my concerns here and elaborate on them in the “Questions” section below.
- A few technical decisions in the methodology and their implications are unclear to me. See my questions below.
- I feel that implementing the proposed model seems quite involved since it requires node resampling and remeshing with Delaunay triangulation.
- I also think the experiments can be improved. I didn’t find too much discussion about its comparisons with the numerical simulators (ANSYS, COMSOL, etc.) or about the method’s generalizability, e.g., to unseen meshes.

**Questions:**

**Questions about flexible dynamics**
- Table 1: Are M, D, T constant matrices or do they have a dependency on x (esp. the stiffness matrix) and/or x_dot (esp. the damping matrix)?
- Table 6: How long did it take ANSYS/COMSOL/etc to simulate these scenes?

**Questions about the mesh edges**
- Is the rest shape information stored as edge features?

**Questions about the contact edge set**
- As the contact is detected on edges/nodes, it looks like the method cannot handle vertex-face collisions. I am worried that this may lead to artifacts in 3D examples.
- I am trying to understand when this set C is computed. Is it dynamically computed for the mesh M^t at every time step, and different time steps may have different C sets? On a related note, could you clarify in what sense C is “fixed” in the description “...C to denote the fixed set of additional contact edges”?

**Questions about node sampling**
- Resampling nodes and re-meshing via Delaunay triangulation seem to be discrete operators. How did backpropagation work via these non-differentiable operators when training the network using Eqn. (8)?

**Questions for node features**
- Isn’t Poisson’s ratio part of the node feature?
- Physical parameters like density and Young’s modulus are stored as node features in this method. Did any examples in your paper use this flexible representation to model spatially varying materials?

**Questions about the experiments**
- C.1 Impact Plate: Is it a 2D or 3D experiment? The setup seems to be a (symmetric) 3D scene, but the text “Therefore, if a ball falls at the center of the plate, there is no need to use a 3D model” seems to imply that it is solved as a 2D example in the approach.
- C1 Impact Plate: “along with 200 validation and test datasets.” Is “datasets” a typo for “trajectories”?
- Table 6: 0.002ms is extremely small. The table indicates that the proposed method is tested on scenes with time steps varying among three orders of magnitude (0.002 vs 10)...Just want to make sure I read it right.

**Minor questions**
- Table 1 caption: should the stress notation be replaced with \sigma so that it is consistent with that in the table?
- The paper repetitively draws comparisons with fluid problems, e.g., EAGLE in Sec. 3.4 and “Notably, the contact edge feature is a novel addition not found in fluid models.” But I don’t get the motivation for such comparisons. The two physical systems are quite different in many ways.

---

> ### Author Response · Authors · 2023-11-17
>
> Thanks for your time reading our paper and leaving insightful comments. We highlight our strengths in the following points.
>
> 1. The long-distance collision response with a hierarchical network architecture is a reasonable idea
> 2. Novelty architecture to capture contact dynamics
>
> Below, we carefully address your concerns. We uploaded our revised paper, where changes are highlighted in blue.
>
> ---
>
> **Q1. Table 1: Are M, D, T constant matrices or do they have a dependency on x (esp. the stiffness matrix) and/or x_dot (esp. the damping matrix)?**
>
> Datasets consist of multiple trajectories obtained from either linear systems (i.e., constant $M,D$, and $T$) or nonlinear systems (i.e., $M(x),D(x)$, and $T(x)$). In the previous presentation, we used the simplified notations $M,D$ and $T$ to avoid notational overload. In the new version, we make sure that we use the general notations (i.e., $M(x),D(x)$, and $T(x)$) which include both the cases (linear and nonlinear).
>
>
> **Q2. Table 6: How long did it take ANSYS/COMSOL/etc to simulate these scenes?**
>
> As we have the direct access only to the simulations associated with the Impact Plate dataset, we report the numbers relevant to the dataset only.
>
> The inference time per step for HCMT is 0.081 s/step and 14.28 s/step for ANSYS, with an average of about 2,278 nodes. Therefore, the proposed method exhibits a much faster simulation speed compared to ANSYS. Information related to time comparison has been added to Appendix F and marked in blue.
>
>
> **Q3. Is the rest shape information stored as edge features?**
>
> The reviewer's understanding is correct.
>
>
> **Q4. As the contact is detected on edges/nodes, it looks like the method cannot handle vertex-face collisions. I am worried that this may lead to artifacts in 3D examples.**
>
> The reviewer is correct. Collisions do not always occur between nodes. While having the same concern and considering this as the issue that should be resolved in future work, we observe that with refined meshes, the learned HCMT models are shown to be able to model 3D rigid dynamics accurately (e.g., Figure 19 in Appendix J).
>
>
> **Q5. I am trying to understand when this set C is computed. Is it dynamically computed for the mesh M^t at every time step, and different time steps may have different C sets? On a related note, could you clarify in what sense C is “fixed” in the description “...C to denote the fixed set of additional contact edges?**
>
> Thanks for pointing out! We corrected the phrase. At every time step, set $C$ is calculated only once during the transformation from $M^t$ to graph ($G_0=V_0,E_0,C)$. In Section 3.3, the expression ‘fixed’ was unclear, so it was revised by being written in blue.
>
>
> **Q6. Resampling nodes and re-meshing via Delaunay triangulation seem to be discrete operators. How did backpropagation work via these non-differentiable operators when training the network using Eqn. (8)?**
>
> We would like to clarify that node pooling and re-meshing are included in the computational graph and  there are no learnable parameters. The proposed method creates a new node index set by pooling nodes from $G_0$ and creates a new edge index set by re-meshing through Delaunay triangulation. This step is repeated for each level in the hierarchy, which is considered in the HMT model. Delaunay triangulation uses the Delaunay function built into SciPy.
>
>
> **Q7. Isn’t Poisson’s ratio part of the node feature?**
>
> Poisson's ratio is a nodal feature. The node features included are different for each system and dataset, but in the case of Impact plate, material parameters such as Young's modulus and density are also included for product design. In addition, when conducting analysis in flexible dynamics where time terms are included, density and Young's modulus are essential according to the governing equations.
>
>
> **Q8. Physical parameters like density and Young’s modulus are stored as node features in this method. Did any examples in your paper use this flexible representation to model spatially varying materials?**
>
> Deformable Plate [1] used Poisson's ratio of -1 to 0.5 as a design parameter. However, among existing materials, Poisson's ratio theoretically ranges between 0 and 0.5. A negative Poisson's ratio means that the material contracts when force is applied. Therefore, we took these factors into consideration when creating the Impact Plate dataset. Except for GGNS [1], we have not found any cases where material parameters were used in node features.
>
> > [1] Linkerhägner, Jonas, et al. "Grounding Graph Network Simulators using Physical Sensor Observations." ICLR 2023 Workshop on Physics for Machine Learning. 2023.

---

> ### Author Response · Authors · 2023-11-17
>
> **Q9. C.1 Impact Plate: Is it a 2D or 3D experiment? The setup seems to be a (symmetric) 3D scene, but the text “Therefore, if a ball falls at the center of the plate, there is no need to use a 3D model” seems to imply that it is solved as a 2D example in the approach.**
>
> The reviewer is correct. As the plate center imposes a symmetric boundary condition, we can leverage 2D analysis, which results in a similar result obtained from 3D analysis, while avoiding intensive computations required in 3D analysis. To clarify this point, the dimensions of each dataset were added to Table 6 and marked in blue.
>
>
> **Q10. C1 Impact Plate: “along with 200 validation and test datasets.” Is “datasets” a typo for “trajectories”?**
>
> Thank you for pointing out! In Appendix C1, “datasets” has been changed to “trajectories” and is marked in blue.
>
>
> **Q11. Table 6: 0.002ms is extremely small. The table indicates that the proposed method is tested on scenes with time steps varying among three orders of magnitude (0.002 vs 10)...Just want to make sure I read it right.**
>
> We would like to confirm that all 50 steps have a time interval of 0.002ms for Impact Plate. The reason for choosing a very short time is to accurately capture the maximum stress of the plate. When the energy of the ball approaches 0, the speed of the ball becomes very slow, so the movement of the ball can be expressed by setting a very short time (when the energy of the ball approaches 0, the maximum stress occurs in the plate).
>
> In addition, at different time-scales, HCMT has been experimentally shown to work well.
>
> As a future researc direction, we believe that solving the problems with step sizes being varied over time will also be an interesting topic.
>
>
> **Q12. Table 1 caption: should the stress notation be replaced with \sigma so that it is consistent with that in the table?**
>
> Thank you for pointing out! In the caption of Table 1, “strain notation” was changed to “stress notation” and written in blue.
>
>
> **Q13. The paper repetitively draws comparisons with fluid problems, e.g., EAGLE in Sec. 3.4 and “Notably, the contact edge feature is a novel addition not found in fluid models.” But I don’t get the motivation for such comparisons. The two physical systems are quite different in many ways.**
>
> We agree that the two physical systems are very different and the expression could be misinterpreted. Our intention was to emphasize that modeling approach for contact dynamics should be different from those of fluid dynamics, motivating us to develop those new features and stressing the importance of contact features in the mesh domain.

---

> ### Author Response · Authors · 2023-11-22
>
> We appreciate the reviewer’s time and effort in reviewing our manuscript and insightful comments. As the closure of the discussion period is approaching, we would like to bring the review’s attention and check if the reviewer could let us know whether the concerns or the misunderstanding have been addressed. If this is the case, we would appreciate if you could adjust your rating accordingly. Thank you!

---

> > ### Comment · Reviewer_zh4D · 2023-11-22
> > **Thank you for the response**
> >
> > Thank you for the clarification and additional data. While I still don't plan to champion this paper, the response has addressed most of my concerns, and I will raise my score to 6 accordingly.

---

### Official Review · Reviewer_J9Rp · 2023-11-01

**Soundness:** 3 good
**Presentation:** 3 good
**Contribution:** 3 good
**Rating:** 6
**Confidence:** 3

**Summary:**

In this paper, the authors introduce the Hierarchical Contact Mesh Transformer (HCMT), a model leveraging hierarchical mesh frameworks to capture long-range dependencies resulting from collisions across spatially distant positions on a body. By using a contact mesh Transformer and a hierarchical mesh Transformer (CMT and HMT) , the HCMT has the ability to quickly propagates collision effects to long range positions. The experiments show that the proposed model can outperform the existing methods on benchmark datasets.

**Strengths:**

- The contact and collision are important and usually challenging problems in physical simulations. The paper aims to address this challenge using transformer for the first time (to my best knowledge). The two transformers architecture i.e., contact mesh Transformer (CMT) for propagating contact messages and hierarchical mesh Transformer (HMT) for handling long-range interactions is intuitive.
- The experiments show that the proposed method can better capture the long-range interaction collision on the test cases provided in both RMSE and visualized results comparing to baselines.

**Weaknesses:**

- The proposed method is only compared with two baseline methods from 2020. There are more recent methods mentioned in the related work of the paper, especially for existing work which some components (e.g. pooling method) of the proposed method are inspired by, are not taken for comparison.
- There are no generalization ability studies for the proposed models. It is unclear how the trained models can be applied to unseen/similar problems without or with minor finetuning.

**Questions:**

- As for the baselines, are there any reasons that only MeshGraphNet (MGN) and Graph Transformer (GT) are selected for comparison?
- How is generalization ability of the proposed models when applied to similar problems? Does it require training from scratch?

---

> ### Author Response · Authors · 2023-11-17
>
> We appreciate Reviewer J9Rp for the insightful comments, highlighting our strengths in the following points.
>
> 1. The paper aims to address contact and collision problems using transformer for the first time.
> 2. Compared to the baseline, RMSE and visualized results show that long-range interaction conflicts can be better captured.
>
> Below we carefully address your concerns. Please refer to the revised paper with changes highlighted in orange and check the [general response](https://openreview.net/forum?id=90yw2uM6J5&noteId=aoJ1BFb5C4).
>
> ---
>
> **Q1. The proposed method is only compared with two baseline methods from 2020. There are more recent methods mentioned in the related work of the paper, especially for existing work which some components (e.g. pooling method) of the proposed method are inspired by, are not taken for comparison. As for the baselines, are there any reasons that only MeshGraphNet (MGN) and Graph Transformer (GT) are selected for comparison?**
>
> We would like to emphasize that recent hierarchical architecture and mesh-based works [1,2,3] focus on fluid dynamics, in which contact does not occur. Morevoer, we found that MS-GNN [1] and GMR-Transformer-GMUS [2] do not have officially released code and we opted against implementing those methods from scratch to avoid potential challenges associated with fair comparisons.
>
> The BSMS-GNN[4] model, which inspired us for the proposed component in our paper, is capable of to predict the position of the Deforming Plate; it did not show better performance than the baseline MGN and we decided not to include them. Here, instead, we conducted additional experiments to verify the effectiveness of the pooling method before and after remesh. The following table shows the results of the evaluations with and without remeshing from Impact Plate. The quality of the mesh was measured using the scaled Jacobian metric. The closer the metric is to 1, the closer it is to an equilateral triangle shape, meaning better quality. We confirmed that the stress RMSE was greatly improved, and it would be good to evaluate it by applying various pooling methods in the future. The results of additional experiments are added to Appendix G.2 and marked in orange.
>
> |       Method      | Mesh Quality(Jacobian) | Position RMSE | Stress RMSE |
> |:-----------------:|:----------------------:|:-------------:|:-----------:|
> | Without remeshing |          0.32          |     19.21     |    17,482   |
> |   With remeshing  |          0.51          |     20.34     |    14,447   |
>
> > [1] Meire Fortunato et al. "MultiScale MeshGraphNets." ICML 2022 2nd AI for Science Workshop. 2022.
> >
> > [2] Xu Han et al. "Predicting Physics in Mesh-reduced Space with Temporal Attention." ICML, 2022.
> >
> > [3] Steeven Janny et al. "EAGLE: Large-Scale Learning of Turbulent Fluid Dynamics with Mesh Transformers." ICLR. 2023.
> >
> > [4] Yadi Cao et al. "Bi-Stride Multi-Scale Graph Neural Network for Mesh-Based Physical Simulation." ICML, 2023
>
>
> **Q2. There are no generalization ability studies for the proposed models. It is unclear how the trained models can be applied to unseen/similar problems without or with minor finetuning. How is generalization ability of the proposed models when applied to similar problems? Does it require training from scratch?**
>
> Please refer to [Q2 in the general response](https://openreview.net/forum?id=90yw2uM6J5&noteId=aoJ1BFb5C4), which describes the experimental results on the proposed model's generalizability.

---

> ### Author Response · Authors · 2023-11-22
>
> We appreciate the reviewer’s time and effort in reviewing our manuscript and insightful comments. As the closure of the discussion period is approaching, we would like to bring the review’s attention and check if the reviewer could let us know whether the concerns or the misunderstanding have been addressed. If this is the case, we would appreciate if you could adjust your rating accordingly. Thank you!

---

> ### Author Response · Authors · 2023-11-23
>
> We believe that we address all the concerns that the reviewer mentioned. We’d greatly appreciate if the reviewer could provide their feedback on our response. Thank you!

---

### Official Review · Reviewer_CgKu · 2023-11-02

**Soundness:** 3 good
**Presentation:** 3 good
**Contribution:** 3 good
**Rating:** 6
**Confidence:** 3

**Summary:**

This manuscript presents a novel mesh-based network architecture, Hierarchical Contact Mesh Transformer, that focuses on flexible body dynamics, which can have large amount of non-linear effects (e.g., collision) in a short time. Using a specially designed mesh pooling layer and a hierarchical architecture, HCMT can do long-range interaction directly and propagate information efficiently. The authors compared HCMT with other baselines using previous datasets and showed that HCMT outperforms them.

**Strengths:**

1. The paper is well written. The presentation and visualization make it an easy one to understand.
2. I personally appreciate the section 3.4 which clarified some confusion when I read there.
3. I appreciate the open-sourced code attached in the manuscript. I believe it will ease the effort when practitioners reproducing this work.
4. The results and comparison to previous methods look impressive and detailed.
5. The ablation study covered a lot of interesting experiments.

**Weaknesses:**

See questions.

**Questions:**

I wonder if this method can enable any downstream tasks that utilize the differentiability of HCMT.

---

> ### Author Response · Authors · 2023-11-17
>
> We appreciate Reviewer CgKu for the insightful comments, highlighting our strengths in the following points.
>
> 1. Clear writing
> 2. Easy to understand the paper via Section 3.4, "Discussion"
> 3. Open-sourced code attached
> 4. Detailed comparison of our proposed model with baselines and impressive results
> 5. The ablation study with interesting experiments
>
> Below, we carefully address your concerns.
>
> ---
>
> **Q1. I wonder if this method can enable any downstream tasks that utilize the differentiability of HCMT.**
>
>
> Since flexible dynamics systems are widely studied in many applications, such as vehicle design, modeling airplane crashes, and mobile phone drop testing, it appears that the proposed modeling approach can be applied to those applications without modifying the core model. The computational analysis obtained from applying the proposed model can be expected to serve as a core technology in the Computer-Aided Engineering (CAE) field. The analysis can also be used in product design because it optimizes virtual designs with computational efficiency (Please refer to [Q1 in the general response](https://openreview.net/forum?id=90yw2uM6J5&noteId=aoJ1BFb5C4)).
>
> Also, as MGN [1] was expanded and applied to several systems, such as static, rigid dynamics, and fluid, we expect that our model can be expanded and applied not only to rigid dynamics including contacts but also to fluid dynamics by utilizing only HMT without CMT.
>
> > [1] Tobias Pfaff, et al. "Learning Mesh-Based Simulation with Graph Networks." ICLR, 2020.

---

> ### Author Response · Authors · 2023-11-22
>
> We appreciate the reviewer’s time and effort in reviewing our manuscript and insightful comments. As the closure of the discussion period is approaching, we would like to bring the review’s attention and check if the reviewer could let us know whether the concerns or the misunderstanding have been addressed. If this is the case, we would appreciate if you could adjust your rating accordingly. Thank you!

---

### Author Response · Authors · 2023-11-17
**Summary of Updates**

We thank all reviewers for their constructive comments. We have uploaded the updated paper accordingly with the following major changes:
- Section 3.3: Update descriptions and the calculation method for contact edges,
- Section 3.3: Update confusing equation,
- Appendix C.1: Add an explanation about Impact Plate,
- Appendix G.2: Add experimental results on whether to remesh,
- Appendix G.3: Add experimental results with clamping,
- Appendix K: Add experimental results of generalization ability,
- Appendix L: Add experimental results of stress singularity experiment.

We highlighted changes in a different color for each reviewer as follows:
- Reviewer gSaT : Red,
- Reviewer zh4D : Blue,
- Reviewer J9Rp : Orange.

---

### Author Response · Authors · 2023-11-17
**General Response to Reviewers**

We thank all reviewers for their valuable time and insightful comments. We make a few general remarks here and respond to individual comments to each review below.

---

**Q1. Importance of Impact Dataset and Its Usability in Industry**

The Impact Plate dataset consists of trajectories of flexible dynamics that are commonly used in the electronic display industry to design display panels (plates). It is mainly used to test the maximum stress occurring in the plate by changing its thickness or material property, typically using commercial tools (e.g., ANSYS, ABAQUS, and so on). This test predicts if the plate will break when the yield strength of material exceeds the stress occurring inside the plate. The result of this test serves as a good indicator for i) checking if the plate is damaged, e.g., by dropping the ball, and ii) allowing the makers to select the optimal thickness and material through virtual simulations.


The dataset includes trajectories collected with varying parameters that are important in the plate design. Plate design parameters include thickness, density, and Young's modulus. In addition, as evaluation conditions, the set of parameters include the ball's drop height and ball size.

We expect benefits in the design optimization for display panels through the utilization of the newly introduced dataset, Impact Plate, and the proposed HCMT model. The HCMT model's capability for accurately modeling the dataset is expected to enhance the efficiency of downstream tasks in the display panel design. Once the experimental conditions (i.e., ball drop height, size) are determined according to the required strength (i.e., energy), an optimal thickness and material that satisfy the strength can be identified. While traditional solvers are computationally demanding, the proposed model can dramatically reduce the time cost for optimization. For detailed results on inference time, please check reviewer zh4D's [Q2](https://openreview.net/forum?id=90yw2uM6J5&noteId=2PY1B908El).


**Q2. Generalization Ability of HCMT**

To test the learned HCMT models' generalizability, we test the models on various design parameters in unseen ranges (i.e., extrapolation in the design parameter space). The table below summarizes the experimental settings and results. Compared to MGN, the proposed MCMT models can achieve accurate predictions on both positions and stress measured in RMSE.

|               Design Parameter              | Parameter range in Training | Parameter range in Evaluating | MGN pos. RMSE | HCMT pos. RMSE | MGN stress RMSE | HCMT stress RMSE |
|:-------------------------------------------:|:---------------------------:|:-----------------------------:|:-------------:|:--------------:|:---------------:|:----------------:|
|             Plate thickness (mm)            |            0.5-1            |             1-1.2             |     54.50     |      17.38     |      25,037     |      12,163      |
|            Ball drop height (mm)            |           500-1000          |           1000-1200           |     31.23     |      29.59     |      35,552     |      22,509      |
| Plate thickness (mm), Ball drop height (mm) |       0.5-1, 500-1000       |        1-1.2, 1000-1200       |     25.23     |      21.52     |      35,552     |      13,745      |

---

### Meta-Review · Area_Chair_i2GA · 2023-12-07

**Metareview:**

The submission deals with physics simulations and proposes a hierarchical method tailored to particularly address the special case of contacts. It has received 4 reviews, which were all borderline but slightly positive. The reviewers appreciated

- the fact that contacts are dealt with, as they are important cases which often do not receive enough attention,
- Detailed experiments and ablation studies,
- good performance.

One of the reviewers suggested that he would raise their score to "7" if this were possible, so the official rating is somewhat lower than the reviewer's aggregated perception.

While the reviewers also raised some issues, in particular wrt to lacking baselines, and lack of generalization, the authors could provide answers, and for the case of generalization provide new experiments. There was also a consensus that the methodological contribution is not strong, but that the paper is still of interest to the field.

The AC concurs and suggests acceptance.

**Justification For Why Not Higher Score:**

The paper could go higher, but there is no real reason given the limited methodological contributions.

**Justification For Why Not Lower Score:**

There is no clear reason to reject this paper.

---

### Decision · Program_Chairs · 2024-01-16

Accept (poster)